# Gradient-Variation Online Learning under Generalized Smoothness

**Yan-Feng Xie, Peng Zhao, Zhi-Hua Zhou**

National Key Laboratory for Novel Software Technology, Nanjing University, China
School of Artificial Intelligence, Nanjing University, China
`{xieyf, zhaop, zhouzh}@lamda.nju.edu.cn`

## Abstract

Gradient-variation online learning aims to achieve regret guarantees that scale with variations in the gradients of online functions, which is crucial for attaining fast convergence in games and robustness in stochastic optimization, hence receiving increased attention. Existing results often require the *smoothness* condition by imposing a fixed bound on gradient Lipschitzness, which may be unrealistic in practice. Recent efforts in neural network optimization suggest a *generalized smoothness* condition, allowing smoothness to correlate with gradient norms. In this paper, we systematically study gradient-variation online learning under generalized smoothness. We extend the classic optimistic mirror descent algorithm to derive gradient-variation regret by analyzing stability over the optimization trajectory and exploiting smoothness locally. Then, we explore *universal online learning*, designing a single algorithm with the optimal gradient-variation regrets for convex and strongly convex functions simultaneously, without requiring prior knowledge of curvature. This algorithm adopts a two-layer structure with a meta-algorithm running over a group of base-learners. To ensure favorable guarantees, we design a new Lipschitz-adaptive meta-algorithm, capable of handling potentially unbounded gradients while ensuring a second-order bound to effectively ensemble the base-learners. Finally, we provide the applications for fast-rate convergence in games and stochastic extended adversarial optimization.

## 1 Introduction

We consider online convex optimization (OCO) [Hazan, 2016; Orabona, 2019], a flexible framework that models the decision-making problem in an online fashion. At each round $t \in [T]$, an online learner is required to submit a decision $\mathbf{x}_t$ from a convex compact set $\mathcal{X} \subseteq \mathbb{R}^d$ and the environments reveal a convex function $f_t : \mathcal{X} \mapsto \mathbb{R}$. Then the learner suffers a loss $f_t(\mathbf{x}_t)$ and updates her decision. The standard performance measure is the *regret* [Zinkevich, 2003] that benchmarks the cumulative loss of the learner against the best decision in hindsight, formally defined as

$$\mathrm{REG}_T = \sum_{t=1}^{T} f_t(\mathbf{x}_t) - \min_{\mathbf{x} \in \mathcal{X}} \sum_{t=1}^{T} f_t(\mathbf{x}). \tag{1}$$

Regret bounds of $\mathcal{O}(\sqrt{T})$ and $\mathcal{O}(\frac{1}{\lambda} \log T)$ are established for convex and $\lambda$-strongly convex functions respectively [Zinkevich, 2003; Hazan et al., 2007]. While these results are known to be minimax optimal [Abernethy et al., 2008], in this paper we are more interested in obtaining *gradient-variation* regret guarantees, which replace the dependence of $T$ in the regret bounds by variations in

---

Correspondence: Peng Zhao <zhaop@lamda.nju.edu.cn>

38th Conference on Neural Information Processing Systems (NeurIPS 2024).

the gradients of online functions [Chiang et al., 2012] defined as

$$V_T = \sum_{t=2}^{T} \sup_{\mathbf{x} \in \mathcal{X}} \|\nabla f_t(\mathbf{x}) - \nabla f_{t-1}(\mathbf{x})\|_2^2. \qquad (2)$$

This quantity can be as small as a constant in stable environments where online functions remain fixed, and is at most $\mathcal{O}(T)$ in the worst case under standard OCO assumptions, safeguarding minimax results. Besides this favorable adaptivity, recent studies have shown close relationships of gradient-variation online learning to various fields, including fast convergence in games [Rakhlin and Sridharan, 2013b; Syrgkanis et al., 2015; Zhang et al., 2022b] and robust stochastic optimization [Sachs et al., 2022; Chen et al., 2024], hence receiving increased attention [Zhao et al., 2020; Yan et al., 2023; Tsai et al., 2023; Ataee Tarzanagh et al., 2024; Zhao et al., 2024].

In online learning, it is proved that the smoothness assumption is necessary for first-order algorithms to achieve gradient-variation regret bounds as discussed in Remark 1 of Yang et al. [2014], which is also restated in Proposition 2 in Appendix B. Previous works typically rely on the *global* $L$-smoothness condition, imposing a fixed upper bound on the gradient Lipschitzness, i.e., requiring $\|\nabla^2 f_t(\mathbf{x})\|_2 \leq L$ for all $t \in [T]$ and $\mathbf{x} \in \mathcal{X}$. However, this global assumption restricts the applicability of theories to loss functions that are quadratically bounded from above. Furthermore, recent studies in neural network optimization have observed phenomena where the global smoothness condition fails to model optimization dynamics effectively, especially for important types of neural networks like LSTM [Zhang et al., 2020b] and Transformer [Crawshaw et al., 2022]. Therefore, modern optimization has devoted efforts to generalizing the smoothness condition. For example, Zhang et al. [2020b] introduce $(L_0, L_1)$-smoothness, which assumes $\|\nabla^2 f(\mathbf{x})\|_2 \leq L_0 + L_1\|\nabla f(\mathbf{x})\|_2$ for an offline objective function $f(\cdot)$. A notable generalization is the recent proposal of the $\ell$-smoothness condition [Li et al., 2023], which assumes $\|\nabla^2 f(\mathbf{x})\|_2 \leq \ell(\|\nabla f(\mathbf{x})\|_2)$ with a link function $\ell(\cdot)$, significantly broadening previous assumptions through the flexibility of $\ell(\cdot)$. Given this, it is natural to ask *how to design online algorithms to exploit generalized smoothness and obtain favorable gradient-variation regret guarantees.*

In this paper, we provide a systematic study of gradient-variation online learning under generalized smoothness. We extend the classic optimistic online mirror descent (optimistic OMD) algorithm [Chiang et al., 2012; Rakhlin and Sridharan, 2013a] to derive gradient-variation regret bounds, achieving $\mathcal{O}(\sqrt{V_T})$ regret and $\mathcal{O}(\log V_T)$ regret for convex and strongly convex functions under generalized smoothness, respectively. We emphasize the importance of stability analysis across the optimization trajectory, which allows generalized smoothness to be effectively exploited locally. Specifically, optimistic OMD maintains two sequences with submitted decisions $\{\mathbf{x}_t\}_{t=1}^T$ and intermediate decisions $\{\widehat{\mathbf{x}}_t\}_{t=1}^T$. We need to control algorithmic stability by appropriate step size tuning and optimism design, ensuring that $\mathbf{x}_t$ is sufficiently close to $\widehat{\mathbf{x}}_t$ to exploit local smoothness at $\widehat{\mathbf{x}}_t$.

Based on this development, we investigate *universal online learning* [van Erven and Koolen, 2016; Wang et al., 2019; Mhammedi et al., 2019; Zhang et al., 2022a; Yan et al., 2023; Yang et al., 2024], where the learner aims to design a single algorithm that simultaneously attains the optimal regret for both convex and strongly functions without the prior knowledge of curvature information. For this scenario, a common wisdom is to adopt an *online ensemble* consisting of a meta-base two-layer structure to handle the environmental uncertainty [Zhao et al., 2024], i.e., the unknown curvature of loss functions, where a meta-algorithm is running over a set of base-learners with different configurations. The base-learners are basically the instantiations of the developed variants of optimistic OMD, as mentioned earlier. However, designing the meta-algorithm is non-trivial with new challenges. The first challenge is from the potentially unbounded smoothness, which might lead to unbounded Lipschitz constants as well. This challenge requires the meta-algorithm to be *Lipschitz-adaptive*, adapting to Lipschitzness on the fly. Furthermore, we also expect it to provide a *second-order regret*, technically required when analyzing the ensemble errors, and to enable *predictions with optimism*, thereby producing the gradient variation. The second challenge is the complexity introduced by the combination procedure inherent in the ensemble method, which further complicates the smoothness estimation, making it difficult to properly tune the meta-algorithm and exploit smoothness.

To this end, we address both challenges with the *function-variation-to-gradient-variation* conversion and a newly-designed Lipschitz-adaptive meta-algorithm. The conversion technique, drawing inspiration from Bai et al. [2022], decouples the design between the meta and base levels and derives the gradient variation directly from function values, allowing us to avoid the cancellation-based analysis [Yan et al., 2023] for utilizing smoothness at the meta level. Nevertheless, this conversion

requires the meta-algorithm to handle *heterogeneous* inputs due to certain technical considerations, and we are not aware of available algorithms satisfying all the requirements, motivating us to design a new algorithm. Based on optimistic Adapt-ML-Prod [Wei et al., 2016] and the clipping technique [Cutkosky, 2019], we present a new Lipschitz-adaptive meta-algorithm with a simpler algorithmic design, which can be of independent interest. With this algorithm, we can apply the function-variation-to-gradient-variation conversion to achieve the optimal results for both convex and strongly convex functions, up to doubly logarithmic factors of $T$, without knowing curvature.

Our findings for gradient-variation online learning are useful for several important applications, including fast-convergence online games [Rakhlin and Sridharan, 2013a; Syrgkanis et al., 2015] and stochastic extended adversarial online learning [Sachs et al., 2022], where we establish new results under the generalized smoothness condition.

The rest of paper is organized as follows. Section 2 provides preliminaries and key ideas for exploiting the generalized smoothness throughout the trajectory. In Section 3 we study universal online learning and present our key meta-algorithm. Section 4 discusses our applications. Related work is provided in Appendix A. All proofs can be found in the remaining appendices (Appendix B – D).

## 2  Gradient-Variation Online Learning under Generalized Smoothness

In this section, we first introduce the problem setup, including the formal definition of generalized smoothness and other assumptions used in the paper. We then extend the optimistic online mirror descent framework to achieve gradient-variation regret bounds under generalized smoothness.

### 2.1  Problem Setup: Generalized Smoothness and Assumptions

Recent studies [Zhang et al., 2020b; Chen et al., 2023b] extend the global smoothness condition by allowing the smoothness to positively correlate with the gradient norm, where a particular function is required to model this relationship. Zhang et al. [2020b] introduce the $(L_0, L_1)$-smoothness condition, where the smoothness is upper bounded by a linear function of the gradient norm, i.e., $\|\nabla^2 f(\mathbf{x})\|_2 \le L_0 + L_1 \|\nabla f(\mathbf{x})\|_2$. Li et al. [2023] further generalize this by imposing a weaker assumption on the link function and propose the *generalized smoothness* defined as follows.

**Definition 1** ($\ell$-smoothness)**.** A twice-differentiable function $f : \mathcal{X} \mapsto \mathbb{R}$ is called $\ell$-smooth for some non-decreasing continuous link function $\ell : [0, +\infty) \mapsto (0, +\infty)$ if it satisfies that $\|\nabla^2 f(\mathbf{x})\|_2 \le \ell(\|\nabla f(\mathbf{x})\|_2)$ for any $\mathbf{x} \in \mathcal{X}$.

The mild requirement on the link function $\ell(\cdot)$ allows for considerable generality. By selecting a linear link function, $\ell$-smoothness immediately recovers $(L_0, L_1)$-smoothness [Zhang et al., 2020b]. Furthermore, it has been shown that $\ell$-smoothness can imply a wide class of functions including rational, logarithmic, and self-concordant functions [Li et al., 2023]. Based on this generalized smoothness notion, we now provide the formal assumption on the smoothness of online functions.

**Assumption 1** (generalized smoothness)**.** The online function $f_t : \mathcal{X} \mapsto \mathbb{R}$ is $\ell_t$-smooth in an open set containing $\mathcal{X} \subseteq \mathbb{R}^d$ for $t \in [T]$, and the learner can query $\ell_t(\mathbf{x})$ provided any point $\mathbf{x} \in \mathcal{X}$.

We also require a standard bounded domain assumption in the OCO literature [Hazan, 2016].

**Assumption 2** (bounded domain)**.** The feasible domain $\mathcal{X} \subseteq \mathbb{R}^d$, which contains the origin $\mathbf{0}$, is non-empty and closed with the diameter bounded by $D$, i.e., $\|\mathbf{x} - \mathbf{y}\|_2 \le D$ for any $\mathbf{x}, \mathbf{y} \in \mathcal{X}$.

We do not assume the prior knowledge of the Lipschitz constant of online functions. In fact, the unboundedness of smoothness may result in unbounded Lipschitz constants. If a Lipschitz upper bound were known, the generalized smoothness condition would be trivialized, as it would allow us to directly compute the upper bound of the smoothness constant. Furthermore, following the discussion in Jacobsen and Cutkosky [2023, Page 2, second paragraph on the right], we assume that there exist finite but *unknown* upper bounds $G$ and $L$ for Lipschitzness and smoothness to ensure the theoretical results are valid. Note that these quantities will only appear in the final regret bounds, and our algorithms does not use them as the inputs. Throughout the paper, we use the $\mathcal{O}(\cdot)$-notation to hide the constants and use the $\widetilde{\mathcal{O}}(\cdot)$-notation to omit the poly-logarithmic factors in $T$.

## 2.2 Algorithmic Framework

We choose optimistic online mirror descent (optimistic OMD) [Rakhlin and Sridharan, 2013a] as the algorithmic framework, which provides a unified view to design and analyze many online algorithms. Compared to classic OMD [Nemirovskij and Yudin, 1985; Beck and Teboulle, 2003], optimistic OMD predicts with side information, an optimistic vector $M_t \in \mathbb{R}^d$. This optimistic vector, also known as optimism, serves as a prediction of the incoming function $f_{t+1}(\cdot)$, leading to tighter regret bounds when accurate. Optimistic OMD updates the decisions in two steps:

$$\mathbf{x}_t = \arg\min_{\mathbf{x} \in \mathcal{X}} \left\{ \langle M_t, \mathbf{x} \rangle + \mathcal{D}_{\psi_t}(\mathbf{x}, \widehat{\mathbf{x}}_t) \right\}, \quad \widehat{\mathbf{x}}_{t+1} = \arg\min_{\mathbf{x} \in \mathcal{X}} \left\{ \langle \nabla f_t(\mathbf{x}_t), \mathbf{x} \rangle + \mathcal{D}_{\psi_t}(\mathbf{x}, \widehat{\mathbf{x}}_t) \right\}, \quad (3)$$

where $\mathcal{D}_{\psi_t}(\mathbf{x}, \mathbf{y}) = \psi_t(\mathbf{x}) - \psi_t(\mathbf{y}) - \langle \nabla \psi_t(\mathbf{y}), \mathbf{x} - \mathbf{y} \rangle$ is the Bregman divergence associated with the regularizer $\psi_t : \mathcal{X} \mapsto \mathbb{R}$. Optimistic OMD maintains two sequences: the sequence of submitted decisions $\{\mathbf{x}_t\}_{t=1}^T$, and the one of intermediate decisions $\{\widehat{\mathbf{x}}_t\}_{t=1}^T$. Although a simplified optimistic OMD with one-step update per round exists [Joulani et al., 2020], we will demonstrate later that tuning the step size based on intermediate decisions is crucial for adapting to generalized smoothness.

## 2.3 Gradient-Variation Regret for Convex and Strongly Convex Functions

When minimizing the convex or strongly convex functions, we set the regularizer as $\psi_t(\mathbf{x}) = \frac{1}{2\eta_t} \|\mathbf{x}\|_2^2$ and optimistic OMD updates with the following steps:

$$\mathbf{x}_t = \Pi_{\mathcal{X}} \left[ \widehat{\mathbf{x}}_t - \eta_t M_t \right], \quad \widehat{\mathbf{x}}_{t+1} = \Pi_{\mathcal{X}} \left[ \widehat{\mathbf{x}}_t - \eta_t \nabla f_t(\mathbf{x}_t) \right], \quad (4)$$

where $\Pi_{\mathcal{X}}[\mathbf{y}] = \arg\min_{\mathbf{x} \in \mathcal{X}} \|\mathbf{x} - \mathbf{y}\|_2$ denotes the Euclidean projection operator. Next, we briefly review approaches for obtaining the gradient-variation bound under *global smoothness*. This bound typically follows from the regret analysis for optimistic OMD:

$$\text{REG}_T \lesssim \frac{1}{\eta_T} + \sum_{t=1}^T \eta_t \|\nabla f_t(\mathbf{x}_t) - M_t\|_2^2 - \sum_{t=1}^T \frac{1}{\eta_t} (\|\mathbf{x}_t - \widehat{\mathbf{x}}_t\|_2^2 + \|\widehat{\mathbf{x}}_t - \mathbf{x}_{t-1}\|_2^2). \quad (5)$$

On the right-hand side, the second term is known as the stability term, while the third one is the negative terms that can be further bounded by $\mathcal{O}\left(-\sum_{t=1}^T \frac{1}{\eta_t} \|\mathbf{x}_t - \mathbf{x}_{t-1}\|_2^2\right)$. Previous studies [Chiang et al., 2012; Zhao et al., 2024] for gradient-variation regret under global smoothness often set optimism as $M_t = \nabla f_{t-1}(\mathbf{x}_{t-1})$, such that, the positive stability term can be upper bounded by $\eta_t \|\nabla f_t(\mathbf{x}_t) - \nabla f_{t-1}(\mathbf{x}_t)\|_2^2 + \eta_t \|\nabla f_{t-1}(\mathbf{x}_t) - \nabla f_{t-1}(\mathbf{x}_{t-1})\|_2^2$, where the first part can be directly converted to the desired gradient variation and the second part will be at most $\eta_t L^2 \|\mathbf{x}_t - \mathbf{x}_{t-1}\|_2^2$ under global smoothness. Given the smoothness constant $L$, tuning the step size as $\eta_t \leq 1/(4L)$ ensures $\mathcal{O}(\eta_t L^2 \|\mathbf{x}_t - \mathbf{x}_{t-1}\|_2^2 - \frac{1}{\eta_t} \|\mathbf{x}_t - \mathbf{x}_{t-1}\|_2^2) \leq 0$, thus obtaining the gradient-variation bound.

However, under generalized smoothness, we do not have a global parameter $L$ for setting the step sizes, and the smoothness constants are related to the decisions. To follow the previous approach, optimistic OMD would require the smoothness constant *before* generating $\mathbf{x}_t$ to tune the step size, ensuring that the negative terms are large enough to cancel $\eta_t \|\nabla f_{t-1}(\mathbf{x}_t) - \nabla f_{t-1}(\mathbf{x}_{t-1})\|_2^2$. Nevertheless, the smoothness constant between $\mathbf{x}_t$ and $\mathbf{x}_{t-1}$ can only be evaluated *after* updating to $\mathbf{x}_t$, resulting in a contradiction. Unlike offline optimization, where the function is fixed and smoothness constants can be shown to decrease along the trajectory [Li et al., 2023], in online optimization, the online functions change at each round, preventing the reuse of previous smoothness estimations.

To address this challenge, our key idea is to perform a trajectory-wise analysis and configure the algorithm using estimated smoothness so far. The key technical lemma is the local smoothness property of $\ell_t$-smooth functions [Li et al., 2023], which allows the smoothness constant between two points to be estimated in advance, provided that the two points are close enough.

**Lemma 1** (local smoothness [Li et al., 2023, Lemma 3.3]). *Suppose $f : \mathcal{X} \mapsto \mathbb{R}$ is $\ell$-smooth. For $\forall \mathbf{x}, \mathbf{y} \in \mathcal{X}$ such that $\|\mathbf{x} - \mathbf{y}\|_2 \leq \frac{\|\nabla f(\mathbf{x})\|_2}{\ell_t(2\|\nabla f(\mathbf{x})\|_2)}$, $\|\nabla f(\mathbf{x}) - \nabla f(\mathbf{y})\|_2 \leq \ell(2\|\nabla f(\mathbf{x})\|_2) \cdot \|\mathbf{x} - \mathbf{y}\|_2$.*

Recall that in the update procedures (3) of optimistic OMD, the submitted decision $\mathbf{x}_t$ is updated based on the intermediate decision $\widehat{\mathbf{x}}_t$. Therefore, it is convenient to control their distance and then exploit the local smoothness at point $\widehat{\mathbf{x}}_t$. Specifically, we set optimism $M_t = \nabla f_{t-1}(\widehat{\mathbf{x}}_t)$ and the step size $\eta_t \leq 1/(4\widehat{L}_{t-1})$, where $\widehat{L}_{t-1} = \ell_{t-1}(2\|\nabla f_{t-1}(\widehat{\mathbf{x}}_t)\|_2)$ denotes the locally estimated

smoothness and is used to tune the step size. This configuration leads to $\eta_t \|\nabla f_t(\mathbf{x}_t) - \nabla f_{t-1}(\widehat{\mathbf{x}}_t)\|_2^2$ for the second term in Eq. (5), which can be further upper bounded as

$$\|\nabla f_t(\mathbf{x}_t) - \nabla f_{t-1}(\widehat{\mathbf{x}}_t)\|_2^2 \leq 2\|\nabla f_t(\mathbf{x}_t) - \nabla f_{t-1}(\mathbf{x}_t)\|_2^2 + 2\|\nabla f_{t-1}(\mathbf{x}_t) - \nabla f_{t-1}(\widehat{\mathbf{x}}_t)\|_2^2. \quad (6)$$

The first part is basically the favorable gradient variation, so it suffices to handle the second part. Performing the stability analysis for OMD and noticing the step size setting, it can be verified that $\|\mathbf{x}_t - \widehat{\mathbf{x}}_t\|_2 \leq \eta_t \|\nabla f_{t-1}(\widehat{\mathbf{x}}_t)\|_2 \leq \|\nabla f_{t-1}(\widehat{\mathbf{x}}_t)\|_2/(4\widehat{L}_{t-1})$. This satisfies the criteria for applying Lemma 1 to the $\ell_{t-1}$-smooth function $f_{t-1}(\cdot)$, allowing us to upper bound the second term in (6) by $\mathcal{O}(\widehat{L}_{t-1}^2 \cdot \|\mathbf{x}_t - \widehat{\mathbf{x}}_t\|_2^2)$. We have clipped $\eta_t$ by $1/(4\widehat{L}_{t-1})$, thereby ensuring the negative term is sufficient to cancel out the positive term. Below, we summarize the result for convex functions.

**Theorem 1.** *Under Assumptions 1 - 2 and assuming online functions are convex, we set the optimism as $M_t = \nabla f_{t-1}(\widehat{\mathbf{x}}_t)$ and $f_0(\cdot) = 0$, with step sizes as $\eta_1 = D$ and, for $t \geq 2$,*

$$\eta_t = \min\left\{ \sqrt{\frac{D^2}{1 + \sum_{s=1}^{t-1}\|\nabla f_s(\mathbf{x}_s) - \nabla f_{s-1}(\mathbf{x}_s)\|_2^2}}, \ \min_{s \in [t]} \frac{1}{4\ell_{s-1}(2\|\nabla f_{s-1}(\widehat{\mathbf{x}}_s)\|_2)} \right\}, \quad (7)$$

*optimistic OMD in (4) ensures the following regret bound,*

$$\text{REG}_T \leq \mathcal{O}\left( D\sqrt{V_T} + \widehat{L}_{\max} \cdot D^2 \right),$$

*where $V_T = \sum_{t=2}^{T} \sup_{\mathbf{x} \in \mathcal{X}} \|\nabla f_t(\mathbf{x}) - \nabla f_{t-1}(\mathbf{x})\|_2^2$ measures the gradient variations and $\widehat{L}_{\max} = \max_{t \in [T]} \widehat{L}_t$ is the maximum smoothness constant over the optimization trajectory.*

This result implies a tighter bound in scenarios where the environments change slowly, i.e., $V_T = \mathcal{O}(1)$. Meanwhile, it safeguards the worst-case optimal result since $V_T \leq \mathcal{O}(T)$ holds in all cases. When assuming $\ell_t(\cdot) \leq L$ for $t \in [T]$, the $\ell_t$-smoothness condition degenerates to the classic global $L$-smoothness condition, and our result implies an $\mathcal{O}(\sqrt{V_T} + LD^2)$ bound, which matches the best-known gradient-variation regret bounds with the first-order oracle [Chiang et al., 2012; Yan et al., 2023; Zhao et al., 2024] even in terms of the dependence on $D$ and $L$. Compared to offline optimization, our result depends on $\widehat{L}_{\max}$, the maximum smoothness constant along the trajectory. This dependence arises from the adversarial nature of online learning, where the loss functions chosen in consecutive rounds may differ significantly, making it hopeless to leverage the previous estimates of smoothness to improve the dependence.

We further provide an improved gradient-variation regret bound for *strongly convex* functions, with step size tuning based on recent result under global smoothness [Chen et al., 2024, § 3.4].

**Theorem 2.** *Under Assumptions 1 - 2 and assuming online functions are $\lambda$-strongly convex, we set the optimism as $M_t = \nabla f_{t-1}(\widehat{\mathbf{x}}_t)$, $f_0(\cdot) = 0$, and step sizes as $\eta_1 = 2/\lambda$ and, for $t \geq 2$, $\eta_t = 2/(\lambda t + 16\max_{s \in [t]} \ell_{s-1}(2\|\nabla f_{s-1}(\widehat{\mathbf{x}}_s)\|_2))$, optimistic OMD in (4) ensures the regret bound $\text{REG}_T \leq \mathcal{O}\left(\frac{1}{\lambda}\log V_T + \widehat{L}_{\max} \cdot D^2\right)$, where $\widehat{L}_{\max} = \max_{t \in [T]} \widehat{L}_t$.*

The above theorem requires the knowledge of curvature information $\lambda$. In Section 3, we design a *universal* method to remove this requirement and achieve the optimal guarantees for convex and strongly convex functions simultaneously without knowing $\lambda$. In Appendix B.2, we discuss the challenge to obtain a gradient-variation bound for exp-concave functions under Assumption 1.

## 3  Universal Online Learning under Generalized Smoothness

Classic online learning algorithms require the curvature information of online functions as algorithmic parameters to achieve favorable regret guarantees. However, obtaining these curvature parameters can be difficult in practice. This challenge motivates the recent study of *universal online learning* [van Erven and Koolen, 2016; Cutkosky and Boahen, 2017; Wang et al., 2019; Mhammedi et al., 2019; Zhang et al., 2022a; Yan et al., 2023; Yang et al., 2024], which aims to design a single algorithm that can achieve optimal regrets without knowing the curvature information. In this section, we study universal online learning with gradient-variation regret under generalized smoothness.

## 3.1 Reviewing Related Work and Techniques

We review related work on gradient-variation universal online learning under *global* smoothness [Zhang et al., 2022a; Yan et al., 2023]. To handle the unknown curvature, universal online learning algorithms utilize a two-layer structure, consisting of a meta-algorithm that ensembles a group of base-learners. Each base-learner optimizes functions with a specific convex curvature, while the meta-algorithm is designed to ensure that ensemble errors do not ruin base-learners' guarantees. Denoted by $N$ the number of base-learners, the decision $\mathbf{x}_t = \sum_{i \in [N]} p_{t,i} \mathbf{x}_{t,i}$ submitted by a two-layer structure algorithm comprises two key components: $\boldsymbol{p}_t \in \Delta_N$, the weights provided by the meta-algorithm, and $\mathbf{x}_{t,i}$, the decision of the $i$-th base-learner. The analysis of a universal algorithm begins by decomposing the regret into two parts against any base-learner. In particular, we choose the base-learner with the best performance (the index $i_\star$ is unknown) as the benchmark:

$$\text{REG}_T = \underbrace{\sum_{t=1}^{T} f_t(\mathbf{x}_t) - \sum_{t=1}^{T} f_t(\mathbf{x}_{t,i_\star})}_{\text{META-REG}} + \underbrace{\sum_{t=1}^{T} f_t(\mathbf{x}_{t,i_\star}) - \min_{\mathbf{x} \in \mathcal{X}} \sum_{t=1}^{T} f_t(\mathbf{x})}_{\text{BASE-REG}}, \tag{8}$$

where the first part is the meta-regret, evaluating the meta-algorithm's performance against the best base-learner, and the second part, known as the base-regret, measures the best learner's performance.

Zhang et al. [2022a] advocate for a simple approach by employing the meta-algorithms with second-order regret guarantees, which facilitates the analysis at the meta level. In specific, they use Adapt-ML-Prod [Gaillard et al., 2014] as the meta-algorithm, showing that the meta-regret for strongly convex and exp-concave functions are constants by exploiting the negative terms from convexity. Consider $\lambda$-strongly convex functions as an example and assume the $i_\star$-th base-learner ensures the optimal $\mathcal{O}(\frac{1}{\lambda} \log V_T)$ base-regret. At the meta level, Zhang et al. [2022a] pass the linearized regret $r_{t,i} = \langle \nabla f_t(\mathbf{x}_t), \mathbf{x}_t - \mathbf{x}_{t,i} \rangle$ to the meta-algorithm for each base-learner. By strong convexity and the guarantees of Adapt-ML-Prod, the meta-regret can be bounded by a constant:

$$\text{META-REG} \leq \sum_{t=1}^{T} r_{t,i_\star} - \frac{\lambda}{2} \sum_{t=1}^{T} \|\mathbf{x}_t - \mathbf{x}_{t,i_\star}\|_2^2 \lesssim \sqrt{\sum_{t=1}^{T} r_{t,i_\star}^2} - \frac{\lambda}{2} \sum_{t=1}^{T} \|\mathbf{x}_t - \mathbf{x}_{t,i_\star}\|_2^2 \leq \mathcal{O}(1),$$

where the last inequality follows from $\sqrt{\sum_t \langle \nabla f_t(\mathbf{x}_t), \mathbf{x}_t - \mathbf{x}_{t,i_\star} \rangle^2} \leq \widehat{G}_{\max} \sqrt{\sum_t \|\mathbf{x}_t - \mathbf{x}_{t,i_\star}\|_2^2}$ and is then canceled by the negative terms via the AM-GM inequality. By leveraging the negative terms from strong convexity, the meta-regret can be well-bounded, allowing the overall regret to be dominated by the base-regret, which is then controlled by selecting appropriate base-algorithms.

However, this method is unsuitable for producing the gradient-variation bound for convex functions. To address it, Yan et al. [2023] propose to use a meta-algorithm which ensures an optimistic and second-order regret bound while provides additional negative terms $-\sum_t \|\boldsymbol{p}_t - \boldsymbol{p}_{t-1}\|_1^2$. Besides showing that the meta-regret is a constant for strongly convex and exp-concave functions following the previous approach, with newly designed optimism, Yan et al. [2023] prove that the meta-regret for convex functions can be roughly bounded by:

$$\mathcal{O}\left( \sqrt{V_T} + \sum_{t=1}^{T} \|\mathbf{x}_{t,i_\star} - \mathbf{x}_{t-1,i_\star}\|_2^2 + L^2 \sum_{t=1}^{T} \|\boldsymbol{p}_t - \boldsymbol{p}_{t-1}\|_1^2 + L^2 \sum_{t=1}^{T} \sum_{i=1}^{N} p_{t,i} \|\mathbf{x}_{t,i} - \mathbf{x}_{t-1,i}\|_2^2 \right).$$

The first term is the gradient variation, matching the optimal order of convex functions. Yan et al. [2023] demonstrate that the remaining stability terms can be canceled through the collaboration between the base and meta levels [Zhao et al., 2024] with the prior knowledge of the global smoothness constant $L$, thus obtaining the near-optimal gradient-variation bounds for convex functions as well. Nevertheless, the employed meta-algorithm already has a two-layer structure, resulting in a three-layer structure for the overall algorithm, which is relatively complicated.

## 3.2 Key Challenges and Main Ideas

We aim to design a universal algorithm with the optimal gradient-variation bounds under generalized smoothness, which exhibits two challenges. First, the Lipschitz condition of online functions is unknown to the meta-algorithm, which requires the meta-algorithm to be *Lipschitz-adaptive*, provide a *second-order regret*, and enable *predictions with optimism*. Second, the combination of the

ensemble method further complicates the estimation of smoothness constants, making it challenging to tune algorithms properly and to cancel stability terms as Yan et al. [2023] did.

We tackle the second challenge by utilizing a *function-variation-to-gradient-variation* conversion to derive the gradient-variation bounds at the meta level, drawing inspiration from the development of dynamic regret minimization [Bai et al., 2022]. This conversion technique decouples the meta and base levels, allowing us to avoid cancellation-based analysis. To illustrate, suppose a meta-algorithm ensuring $\mathcal{O}(\sqrt{\sum_t (\ell_{t,i_\star} - m_{t,i_\star})^2})$ provided optimism $\boldsymbol{m}_t = (m_{t,1}, \ldots, m_{t,N})$. By setting $\ell_{t,i_\star} = f_t(\mathbf{x}_{t,i_\star}) - f_t(\mathbf{x}_{\text{ref}})$ and $m_{t,i_\star} = f_{t-1}(\mathbf{x}_{t,i_\star}) - f_{t-1}(\mathbf{x}_{\text{ref}})$, where $\mathbf{x}_{\text{ref}}$ is a fixed reference point, the meta regret bound becomes $\mathcal{O}(\sqrt{\sum_t [(f_t(\mathbf{x}_{t,i}) - f_{t-1}(\mathbf{x}_{t,i})) - (f_t(\mathbf{x}_{\text{ref}}) - f_{t-1}(\mathbf{x}_{\text{ref}}))]^2})$. By the mean value theorem, $[(f_t(\mathbf{x}_{t,i}) - f_{t-1}(\mathbf{x}_{t,i})) - (f_t(\mathbf{x}_{\text{ref}}) - f_{t-1}(\mathbf{x}_{\text{ref}}))]^2$ equals the first term below, which can be further upper bounded by the gradient variation:

$$[\langle \nabla f_t(\xi_{t,i}) - \nabla f_{t-1}(\xi_{t,i}), \mathbf{x}_{t,i} - \mathbf{x}_{\text{ref}} \rangle]^2 \leq D^2 \sup_{\mathbf{x} \in \mathcal{X}} \|\nabla f_t(\mathbf{x}) - \nabla f_{t-1}(\mathbf{x})\|_2^2.$$

This technique brings hope for minimizing the convex functions. To develop a universal method, our first attempt is to utilize MsMwC-Master [Chen et al., 2021] as the meta-algorithm, which satisfies all the three requirements imposed by the first challenge. Nevertheless, the *heterogeneous* inputs at the meta level present another challenge to this approach. The heterogeneity arises as we pass $r_{t,i} = f_t(\mathbf{x}_t) - f_t(\mathbf{x}_{t,i})$ to the meta-algorithm for base-learner responsible for convex functions, leveraging the conversion technique, while $r_{t,i} = \langle \nabla f_t(\mathbf{x}_t), \mathbf{x}_t - \mathbf{x}_{t,i} \rangle$ for base-learners minimizing strongly convex functions. It remains unclear how to adapt MsMwC-Master [Chen et al., 2021] to our heterogeneous inputs, as MsMwC-Master requires an $\ell_t$ as inputs, and the guarantee is for the regret in the form of $\boldsymbol{r}_t = \langle \boldsymbol{p}_t, \ell_t \rangle - \ell_t$. However, such an $\ell_t$ cannot be retrieved from our above design. Fortunately, we observe that the Prod algorithms [Cesa-Bianchi et al., 2007; Gaillard et al., 2014; Wei et al., 2016] are friendly to heterogeneous inputs. Technically, the Prod algorithms provide the same guarantees as long as $\sum_{i \in [N]} p_{t,i} r_{t,i} \leq 0$, thanks to the potential-based analysis. Therefore, aside from the requirements of the first challenge, we expect that the meta-algorithm can be analyzed similarly to the Prod algorithms, motivating us to design a new meta-algorithm. As a byproduct, we present a universal algorithm with a two-layer structure under global smoothness with the developed techniques. It is more efficient than that by Yan et al. [2023] and attains the optimal gradient-variation guarantees for convex, strongly convex, and exp-concave functions, at a cost of additional function value queries. We defer algorithms and regret bounds to Appendix C.5.

### 3.3 A New Lipschitz-Adaptive Meta-Algorithm

In Algorithm 1, we present our meta-algorithm, which builds on optimistic Adapt-ML-Prod [Wei et al., 2016] and incorporates the clipping technique introduced by Cutkosky [2019] and further refined by Chen et al. [2021]. This algorithm, described in the language of Prediction with Experts' Advice (PEA), may be of independent interest beyond adapting to the generalized smoothness. Apart for satisfying all expected requirements, this algorithm offers a simpler design, which does not require a forced restart as opposed to MsMwC-Master [Chen et al., 2021].

This efficiency improvement is achieved through a simple self-confident learning rate in Line 6 of Algorithm 1, unlike that uses fixed learning rates and thus require restarts. In essence, our approach incorporates the clipping mechanism by adding $B_t^2$ to the denominator and removing the threshold on learning rates commonly applied in prior Prod algorithms [Gaillard et al., 2014; Wei et al., 2016]. This term $B_t^2$ acts as a threshold, ensuring that $\eta_{t,i}|\bar{r}_{t,i} - m_{t,i}| \leq 1/2$, a critical condition in the analysis (typically satisfied when the Lipschitz constant is provided for prior Prod algorithms). Lipschitz-adaptive algorithms may be sensitive to the choice of $B_0$, while in our case, $B_0 = \Theta(1/(\log T))$ is sufficient and does not ruin the guarantees as the factor $\mathcal{O}(\log\log)$ is often treated as a constant [Gaillard et al., 2014; Luo and Schapire, 2015]. Theorem 3 summarizes the guarantee of Algorithm 1 and the proof is provided in Appendix C.3.

**Theorem 3.** *Setting* $m_{t,i} = \langle \boldsymbol{p}_t, \ell_{t-1} \rangle - \ell_{t-1,i}$ *in Algorithm 1 ensures that, for any* $i_\star \in [N]$, $\sum_{t=1}^T \langle \boldsymbol{p}_t, \ell_t \rangle - \sum_{t=1}^T \ell_{t,i_\star}$ *is bounded as follows, where* $B_T = \max\{B_0, \max_{t \in [T]} \|\boldsymbol{r}_t - \boldsymbol{m}_t\|_\infty\}$:

$$\mathcal{O}\left(\left(\sqrt{\sum_{t=1}^T (r_{t,i_\star} - m_{t,i_\star})^2} + B_T\right) \cdot \left(\log(N) + \log(B_T + \log T)\right)\right) \leq \widetilde{\mathcal{O}}\left(\sqrt{\sum_{t=1}^T \|\ell_t - \ell_{t-1}\|_\infty^2}\right).$$

---

**Algorithm 1** Lipschitz Adaptive Optimistic Adapt-ML-Prod

---

**Input:** prior information of the scale $B_0$, the number of experts $N$.

1: **Initialization:** set $w_{1,i} = 1$, $m_{1,i} = 0$ and $\eta_{1,i} = 1/\sqrt{1 + 4B_0^2}$ for all $i \in [N]$.
2: **for** $t = 1$ **to** $T$ **do**
3:     Update the weight for $i \in [N]$ $\widetilde{w}_{t,i} = w_{t,i} \exp(\eta_{t,i} m_{t,i})$;
4:     Calculate decision $\boldsymbol{p}_t \in \Delta_N$ with $p_{t,i} = \frac{\eta_{t,i} \widetilde{w}_{t,i}}{\sum_{j \in [N]} \eta_{t,j} \widetilde{w}_{t,j}}$ and submit it;
5:     Receive $\boldsymbol{r}_t$, update $B_t = \max\{B_{t-1}, \|\boldsymbol{r}_t - \boldsymbol{m}_t\|_\infty\}$, and build $\bar{r}_{t,i} = m_{t,i} + \frac{B_{t-1}}{B_t}(r_{t,i} - m_{t,i})$;
6:     Update the learning rate for $i \in [N]$: $\eta_{t+1,i} = \sqrt{\frac{1}{1 + \sum_{s=1}^{t}(\bar{r}_{s,i} - m_{s,i})^2 + 4B_t^2}}$;
7:     Update the weight for $i \in [N]$: $w_{t+1,i} = \left(w_{t,i} \exp\left(\eta_{t,i}\bar{r}_{t,i} - \eta_{t,i}^2(\bar{r}_{t,i} - m_{t,i})^2\right)\right)^{\frac{\eta_{t+1,i}}{\eta_{t,i}}}$.
8: **end for**

---

**Algorithm 2** Universal Gradient-Variation Online Learning under Generalized Smoothness

---

**Input:** curvature coefficient pool $\mathcal{H}$, number of base-learners $N$, prior information of the scale $B_0$.

1: **Initialization**: Send $N$ and $B_0$ to the meta-algorithm, for $\lambda \in \mathcal{H}$, initialize an algorithm in Theorem 2 with it; initialize an algorithm in Theorem 1.
2: **for** $t = 1$ **to** $T$ **do**
3:     Obtain $\boldsymbol{p}_t$ from meta-algorithm, $\mathbf{x}_{t,i}$ from each base-learner $i \in [N]$;
4:     Submit $\mathbf{x}_t = \sum_{i \in [N]} p_{t,i} \mathbf{x}_{t,i}$;
5:     Receive $f_t(\cdot)$ and send it to each base-learner for update;
6:     **For strongly convex functions learners**: set $r_{t,i} = \langle \nabla f_t(\mathbf{x}_t), \mathbf{x}_t - \mathbf{x}_{t,i} \rangle$;
7:     **For convex functions learner**: set $r_{t,i} = f_t(\mathbf{x}_t) - f_t(\mathbf{x}_{t,i})$;
8:     Send $m_{t+1,i} = f_t(\mathbf{x}_t) - f_t(\mathbf{x}_{t,i})$ to the meta-algorithm for $i \in [N]$.
9: **end for**

---

Our algorithm improves efficiency at the cost of an additional factor $\mathcal{O}(\sqrt{\log N})$. This factor is ignorable for universal online learning since we set $N = \mathcal{O}(\log T)$, and the factor $\mathcal{O}(\log \log T)$ is negligible. Considering other related Lipschitz-adaptive algorithms, Mhammedi et al. [2019] obtain a regret bound of $\mathcal{O}(\sqrt{\sum_t (r_{t,i_\star})^2 \cdot (\log(N) + \log \log(B_T T))} + B_T \log(N))$, which offers better dependence on the dominant term $\sqrt{\sum_t (r_{t,i_\star})^2}$ but it is unclear how to include optimism. Chen et al. [2021] achieve a bound of $\mathcal{O}(\sqrt{\sum_t (\ell_{t,i_\star} - m_{t,i_\star})^2 \cdot \log(NT)} + B_T \log(NT))$ with a two-layer algorithm; however, the $\mathcal{O}(\sqrt{\log T})$ term would ruin the desired $\mathcal{O}(\log V_T)$ bound for strongly convex functions. We remark that the compared methods enjoy other strengths not discussed here, such as the ability to compete with an arbitrary competitor $\boldsymbol{x} \in \Delta_N$ and the versatility to handle various learning scenarios, while our method is sufficient for our purpose and the only option to tackle all the challenges as we mention in Section 3.2. Lastly, notice that the optimism $\boldsymbol{m}_t$ involves the decision $\boldsymbol{p}_t$, which might be improper since $\boldsymbol{p}_t$ depends on $\boldsymbol{m}_t$ as well. We refer readers to Appendix C.1 for efficiently setting $\boldsymbol{m}_t$ through a univariate binary search.

We emphasize that optimism $\boldsymbol{m}_t$ in our algorithm is not chosen arbitrarily. In Line 5, we clip the regret with optimism to keep them on the same scale. The performance is then evaluated based on the clipped regret $\bar{r}_{t,i}$. For the analytical purpose, it is essential that $\langle \boldsymbol{p}_t, \bar{\boldsymbol{r}}_t \rangle \leq 0$, and a sufficient condition for this is ensuring $\langle \boldsymbol{p}_t, \boldsymbol{m}_t \rangle \leq 0$, which imposes an additional requirement on $\boldsymbol{m}_t$. In Appendix C.2, we discuss how this requirement introduces challenges for exp-concave functions minimization in universal online learning.

## 3.4 Overall Algorithm and Regret Guarantees

The function-variation-to-gradient-variation technique decouples the design of universal methods into the base and meta levels, and we are ready to combine the proposed components together. We employ algorithms in Section 2.3 as the base-learners and use Algorithm 1 as the meta-learner, concluding in Algorithm 2. Theorem 4 presents its guarantee with the proof in Appendix C.4.

**Theorem 4.** *Under Assumptions 1 - 2 and assuming a global lower bound such that $\underline{f} \leq f_t(\mathbf{x})$ for any $\mathbf{x} \in \mathcal{X}, t \in [T]$, setting $N = \lceil \log_2 T \rceil + 1$, defining the curvature coefficient pool $\mathcal{H} = $*

$\{2^{i-1}/T : i \in [N-1]\}$, and specifying $B_0$, Algorithm 2 simultaneously ensures:

$$\text{REG}_T \leq \begin{cases} \mathcal{O}(\sqrt{V_T} \cdot \log B_T), & \textit{(convex)}, \\ \mathcal{O}\left(\frac{1}{\lambda} \log V_T + \widehat{G}_{\max}^2 \log^2(B_T)/\lambda + B_T \log B_T\right), & (\lambda\textit{-strongly convex}), \end{cases}$$

where $\lambda \in [1/T, 1]$, $B_T = \mathcal{O}(\max\{B_0, D \max_{t \in [T]} \sup_{\mathbf{x}} \|\nabla f_t(\mathbf{x}) - \nabla f_{t-1}(\mathbf{x})\|_2\})$ and $\widehat{G}_{\max}$ is the Lipschitz constant on the optimization trajectory.

Without loss of generality, we assume $\lambda \in [1/T, 1]$ for strongly convex functions. If $\lambda < 1/T$, the optimal $\mathcal{O}((\log V_T)/\lambda)$ bound would imply linear regret, in which case we would treat them as general convex functions. If $\lambda > 1$, our result is slightly worse than the optimal one by a negligible constant factor. This simplification is also employed by Zhang et al. [2022a]; Yan et al. [2023].

**Remark 1.** This theorem additionally requires the lower bound of loss functions, which is used to perform the binary search when setting the optimism. We defer the details of the binary search to Appendix C.1. This assumption is also employed recently in parameter-free optimizations [Hazan and Kakade, 2019; Attia and Koren, 2024; Khaled and Jin, 2024], and we can simply choose $\underline{f} = 0$ in empirical risk minimization settings [Hazan and Kakade, 2019].

## 4 Applications

In this section, we demonstrate the importance of our results by providing two applications (SEA model and online games), where new results can be directly implied from our findings.

### 4.1 Stochastically Extended Adversarial (SEA) Model

The stochastically extended adversarial (SEA) model [Sachs et al., 2022] interpolates adversarial and stochastic online optimization. It assumes that the environments select the loss function $f_t(\cdot)$ from a distribution $\mathfrak{D}_t$. The adversarial nature is characterized by shifts in distribution $\mathfrak{D}_t$, and when $\mathfrak{D}_t$ remains constant, the model captures the environments' stochastic behavior. The following quantities are introduced to measure the levels of adversarial and stochastic behaviors in environments:

$$\Sigma_{1:T}^2 = \mathbb{E}\left[\sum_{t=2}^{T} \sup_{\mathbf{x} \in \mathcal{X}} \|\nabla F_t(\mathbf{x}) - \nabla F_{t-1}(\mathbf{x})\|_2^2\right], \sigma_{1:T}^2 = \sum_{t=1}^{T} \sup_{\mathbf{x} \in \mathcal{X}} \mathbb{E}[\|\nabla f_t(\mathbf{x}) - \nabla F_t(\mathbf{x})\|_2^2]. \quad (9)$$

where we denote by $F_t(\mathbf{x}) = \mathbb{E}_{f_t \sim \mathfrak{D}_t}[f_t(\mathbf{x})]$. In above, $\Sigma_{1:T}^2$ represents the adversarial shift of the distribution, and $\sigma_{1:T}^2$ denotes the stochastic variance.

Sachs et al. [2022] prove an $\mathcal{O}(\sqrt{\sigma_{1:T}^2} + \sqrt{\Sigma_{1:T}^2})$ regret for convex functions, and a refined regret bound of $\mathcal{O}((\sigma_{\max}^2 + \Sigma_{\max}^2) \log(\sigma_{1:T}^2 + \Sigma_{1:T}^2))$ for strongly convex functions is obtained by Chen et al. [2023a]; Sachs et al. [2023], where $\sigma_{\max}^2 = \max_{t \in [T]} \sup_{\mathbf{x} \in \mathcal{X}} \mathbb{E}[\|\nabla f_t(\mathbf{x}) - \nabla F_t(\mathbf{x})\|_2^2]$ and $\Sigma_{\max}^2 = \max_{t=1}^{T} \sup_{\mathbf{x} \in \mathcal{X}} \|\nabla F_t(\mathbf{x}) - \nabla F_{t-1}(\mathbf{x})\|_2^2$. Yan et al. [2023] present a universal method which can obtain $\widetilde{\mathcal{O}}(\sqrt{\sigma_{1:T}^2} + \sqrt{\Sigma_{1:T}^2})$ and $\mathcal{O}((\sigma_{\max}^2 + \Sigma_{\max}^2) \log(\sigma_{1:T}^2 + \Sigma_{1:T}^2))$ bounds for convex and strongly convex functions. However, these results require the global smoothness assumption.

Our result in Section 3 implies a new finding for the SEA model, relaxing the assumption from the global smoothness to the generalized smoothness, while adapting to unknown curvature, summarized in Corollary 1. The proof can be found in Appendix D.1.

**Corollary 1.** *Under Assumptions 1 - 2 and assuming a global lower bound for the loss functions such that $\underline{f} \leq f_t(\mathbf{x})$ for any $\mathbf{x} \in \mathcal{X}, t \in [T]$, setting $N = \lceil \log_2 T \rceil + 1$, defining the curvature coefficient pool $\mathcal{H} = \{2^{i-1}/T : i \in [N-1]\}$, and specifying $B_0$ with a specific value, then, under the SEA model, Algorithm 2 simultaneously ensures:*

$$\mathbb{E}[\text{REG}_T] \leq \begin{cases} \mathcal{O}\big((\sqrt{\widetilde{\sigma}_{1:T}^2} + \sqrt{\Sigma_{1:T}^2}) \cdot \log(\widehat{G}_{\max} D)\big), & \textit{(convex)}, \\ \mathcal{O}\left((\widetilde{\sigma}_{\max}^2 + \Sigma_{\max}^2) \log(\widetilde{\sigma}_{1:T}^2 + \Sigma_{1:T}^2)\right), & \textit{(strongly convex)}, \end{cases}$$

*where $\widehat{G}_{\max}$ is the maximum empirical Lipschitz constant, $\widetilde{\sigma}_{1:T}^2 = \sum_{t=1}^{T} \mathbb{E}[\sup_{\mathbf{x} \in \mathcal{X}} \|\nabla f_t(\mathbf{x}) - \nabla F_t(\mathbf{x})\|_2^2]$, and $\widetilde{\sigma}_{\max}^2 = \mathbb{E}[\max_{t \in [T]} \sup_{\mathbf{x} \in \mathcal{X}} \|\nabla f_t(\mathbf{x}) - \nabla F_t(\mathbf{x})\|_2^2]$.*

Note that, in real-world streaming learning applications, this corollary can offer a more generalized depiction of data throughput with limited computing resources [Zhou, 2024; Wang et al., 2024], given the connection between these scenarios and the SEA model [Chen et al., 2024, § 5.6]. Our result depends on $\widetilde{\sigma}_{1:T}^2$, a larger quantity than $\sigma_{1:T}^2$ but still can track the stochastic variance. This is because our algorithm utilizes the information afterward $\mathbf{x}_{t-1}$ to generate $\mathbf{x}_t$. We refer the interested readers for this subtle issue to the discussion by Chen et al. [2024]. This dependence currently is unknown how to improve even under the global smoothness condition since we need to leverage the function variation to produce gradient variation, inevitably involving the afterward information.

## 4.2 Fast Rates in Games

Our second application explores the min-max game, aiming to achieve an $\varepsilon$-approximate solution to the problem $\min_{\mathbf{x} \in \mathcal{X}} \max_{\mathbf{y} \in \mathcal{Y}} f(\mathbf{x}, \mathbf{y})$ within an $\mathcal{O}(1/T)$ fast convergence rate. Here, we assume that $f(\cdot, \mathbf{y})$ is convex for any $\mathbf{y} \in \mathcal{Y}$, and correspondingly, $f(\mathbf{x}, \cdot)$ is concave given any $\mathbf{x} \in \mathcal{X}$. Additionally, we assume that both $\mathcal{X} \subset \mathbb{R}^n$ and $\mathcal{Y} \subset \mathbb{R}^m$ are bounded convex sets. Pioneering research [Syrgkanis et al., 2015] demonstrates that optimistic algorithms can reach a convergence rate of $\mathcal{O}(1/T)$ by leveraging gradient variation. However, these results are limited to the global smoothness condition. In this part, we demonstrate that our findings in Section 2 directly imply a new algorithm that can exploit generalized smoothness.

Following the notations of Nemirovski [2004], we define $\mathcal{Z} = \mathcal{X} \times \mathcal{Y}$, $\mathbf{z} = (\mathbf{x}, \mathbf{y}) \in \mathcal{Z}$ and introduce an operator $F : \mathcal{Z} \to \mathbb{R}^n \times \mathbb{R}^m$ with $F(\mathbf{z}) = (\nabla_{\mathbf{x}} f(\mathbf{x}, \mathbf{y}), -\nabla_{\mathbf{y}} f(\mathbf{x}, \mathbf{y}))$. We extend the concept of $\ell$-smoothness to the min-max optimization setting as follows.

**Definition 2** ($\ell$-smoothness for min-max game). *A differentiable convex-concave function $f : \mathcal{X} \times \mathcal{Y} \mapsto \mathbb{R}$ is called $\ell$-smooth with a non-decreasing link function $\ell : [0, +\infty) \mapsto (0, +\infty)$ if it satisfies: for any $\mathbf{z}_1, \mathbf{z}_2 \in \mathcal{Z}$, if $\mathbf{z}_1, \mathbf{z}_2 \in \mathbb{B}\big(\mathbf{z}, \frac{\|F(\mathbf{z})\|_2}{\ell(2\|\nabla F(\mathbf{z})\|_2)}\big)$, then $\|F(\mathbf{z}_1) - F(\mathbf{z}_2)\|_2 \leq \ell(2\|F(\mathbf{z})\|) \cdot \|\mathbf{z}_1 - \mathbf{z}_2\|_2$, where $\mathbb{B}(\mathbf{z}, r)$ denotes the Euclidean ball centered at point $\mathbf{z}$ with radius $r$.*

This definition is a counterpart to Definition 1 in the min-max game, but weaker as it does not require the twice-differentiability requirement. The $\varepsilon$-approximate solution $(\mathbf{x}_\star, \mathbf{y}_\star)$ to the min-max game is formally defined by $f(\mathbf{x}_\star, \mathbf{y}) - \varepsilon \leq f(\mathbf{x}_\star, \mathbf{y}_\star) \leq f(\mathbf{x}, \mathbf{y}_\star) + \varepsilon$ for any $\mathbf{x} \in \mathcal{X}, \mathbf{y} \in \mathcal{Y}$. To achieve the fast convergence rate to the solution, indeed our result in Section 2 can be directly applied to the min-max game and obtains the following tailored algorithm for min-max optimization:

$$\mathbf{z}_t = \Pi_{\mathcal{Z}} \left[\widehat{\mathbf{z}}_t - \eta_t F(\widehat{\mathbf{z}}_t)\right], \quad \widehat{\mathbf{z}}_{t+1} = \Pi_{\mathcal{Z}} \left[\widehat{\mathbf{z}}_t - \eta_t F(\mathbf{z}_t)\right]. \tag{10}$$

In above we set the optimism at the point $\widehat{\mathbf{z}}_t$ in order to exploit the smoothness locally. We conclude our result in Corollary 2, with the proof available in Appendix D.2.

**Corollary 2.** *Assume that the convex-concave function $f(\mathbf{x}, \mathbf{y})$ is $\ell$-smooth, and the domain $\mathcal{Z}$ is bounded with diameter $D$. By applying the tuning strategy described in Theorem 1, and defining the final approximated solution as $\bar{\mathbf{z}}_T = \frac{1}{T} \sum_{t=1}^{T} \mathbf{z}_t$, where $\mathbf{z}_t$ is generated by Eq. (10), we achieve an $\varepsilon$-approximate solution with a convergence rate of $\mathcal{O}(1/T)$.*

## 5 Conclusion

In this paper, we provide a systematic study of gradient-variation online learning under the generalized smoothness condition. We exploit trajectory-wise smoothness to achieve the optimal regret bounds: $\mathcal{O}(\sqrt{V_T})$ for convex functions and $\mathcal{O}(\log V_T)$ for strongly convex functions, respectively. We further consider more complicated online learning scenarios, motivating us to design a new Lipschitz-adaptive meta-algorithm, which can be of independent interest. Hinging on this algorithm with the function-variation-to-gradient-variation technique, we design a universal algorithm which guarantees the optimal results for convex functions and strongly convex functions simultaneously without knowing the curvature. In addition, our findings directly imply new results in stochastic extended adversarial online learning and fast-rate games under generalized smoothness.

An important future direction for future research is to explore whether our method can be further extended to accommodate the one-gradient feedback model, where the learner receives only the gradient information of the decision submitted in each round. Another interesting problem is to exploit the exp-concavity in gradient-variation online learning under generalized smoothness.

## Acknowledgements

This research was supported by National Key R&D Program of China (2022ZD0114800), NSFC (U23A20382), and JiangsuSF (BK20220776). Peng Zhao was supported in part by the Xiaomi Foundation.

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

# A   Related Work

In this section, we review the related work in gradient-variation online learning, the generalized smoothness conditions, and the Lipschitz-adaptive algorithms.

## A.1   Gradient-Variation Online Learning

The gradient-variation quantity defined in Eq. (2) is firstly introduced by Chiang et al. [2012] for global smooth functions. Chiang et al. [2012] obtain $\mathcal{O}(\sqrt{V_T})$ and $\mathcal{O}(\frac{d}{\alpha}\log V_T)$ regret bounds respectively for general convex and $\alpha$-exp-concave functions. Zhang et al. [2022a] later achieve $\mathcal{O}(\frac{1}{\lambda}\log V_T)$ result for $\lambda$-strongly convex functions. Considering the non-stationary environments, Zhao et al. [2020] study gradient variation under the dynamic regret, a strengthened measure that requires the learner to compete with a sequence of time-varying comparators. Their work revives the study of gradient-variation online learning, especially revealing the importance of the stability analysis in the two-layer online ensemble. Their results are further improved in Zhao et al. [2024], where they only require one gradient per round with the same optimal guarantees through the collaboration between the meta and the base. For universal online learning [van Erven and Koolen, 2016], there are several recent researches trying to derive the gradient-variation bounds in this context [Zhang et al., 2022a; Sachs et al., 2023; Yan et al., 2023].

Gradient variation demonstrates its importance due to its close connection with many online learning problems. Pioneering works [Rakhlin and Sridharan, 2013a; Syrgkanis et al., 2015; Zhang et al., 2022b] demonstrate the importance of gradient variation via a useful property (Regret bounded by Variation in Utilities, RVU) to achieve fast-rate convergences in multi-player games. Recently, Sachs et al. [2022] and Chen et al. [2024] reveal that the gradient variation is also essential in bridging stochastic and adversarial convex optimization.

However, all the mentioned works require the global smoothness assumption. As highlighted in Proposition 2 in Appendix B, the smoothness condition is necessary to obtain the gradient-variation bounds for algorithms with first-order oracle assumption [Nesterov, 2018]. Exceptions are that Jacobsen and Cutkosky [2022]; Bai et al. [2022] obtain the gradient-variation bounds without the smoothness assumption, but they require *implicit updates* [Kulis and Bartlett, 2010; Nicolò and Francesco, 2020] per round, which may be inefficient under online settings. Our work considers generalized smoothness, and we develop first-order methods to achieve gradient-variation bounds.

## A.2   Generalized Smoothness

Generalized smoothness has received increasing attention in recent years since the analysis under the standard global smoothness is insufficient to depict the dynamics of neural network optimization. Based on the empirical observation for the relationship between the smoothness and the gradients of LSTMs, Zhang et al. [2020b] relax the global smoothness assumption by $(L_0, L_1)$-smoothness condition, which assumes $\|\nabla^2 f(\mathbf{x})\|_2 \leq L_0 + L_1\|\nabla f(\mathbf{x})\|_2$ for offline objective function $f : \mathcal{X} \mapsto \mathbb{R}$. With this new smoothness assumption, Zhang et al. [2020b] explain the importance of gradient clipping in neural network training. There are a variety of subsequent works developed for different methods and settings [Zhang et al., 2020a; Crawshaw et al., 2022; Reisizadeh et al., 2023; Faw et al., 2023; Wang et al., 2023]. There are studies further generalizing the $(L_0, L_1)$-smoothness condition. Chen et al. [2023b] introduce the $\alpha$-symmetric smoothness condition, which symmetrizes the $(L_0, L_1)$-smoothness condition and allows the smoothness to depend polynomially on the gradient. Notably, Li et al. [2023] propose the $\ell$-smoothness, defined as $\|\nabla^2 f(\mathbf{x})\|_2 \leq \ell(\|\nabla f(\mathbf{x})\|_2)$. This condition does not specify a particular form for the function $\ell : \mathbb{R} \to \mathbb{R}$, other than some mild assumptions about its properties, thereby allowing great generality of this notion.

Telgarsky [2022] introduces the concept of $(G, L)$-quadratically bounded functions, which aims to generalize the Lipschitz condition as $\|\nabla f(\mathbf{x})\|_2 \leq G + L\|\mathbf{x} - \mathbf{x}_0\|_2$ with a reference point $\mathbf{x}_0 \in \mathcal{X}$. Though this notion covers the standard global smoothness condition, it lacks a detailed depiction of the relationship between the smoothness and the gradients, hindering further research into understanding the optimization dynamics such as the role of gradient clipping [Zhang et al., 2020b]. Jacobsen and Cutkosky [2023] investigate online convex optimization under this constraint such that the online functions may be unbounded. However, it is unclear how to obtain the gradient-variation regret in their context and using their methods.

We also mention another generalization of the standard smoothness called the *relative smoothness* [Lu et al., 2018]. A function $f : \mathcal{X} \mapsto \mathbb{R}$ is $L$-smooth relative function if $f(\mathbf{x}) \leq f(\mathbf{y}) + \langle \nabla f(\mathbf{y}), \mathbf{x} - \mathbf{y} \rangle + L\mathcal{D}_h(\mathbf{x}, \mathbf{y})$ for any $\mathbf{x}, \mathbf{y} \in \text{int } \mathcal{X}$, where $h : \mathcal{X} \mapsto \mathbb{R}$ is a reference function with $\mathcal{D}_h(\mathbf{x}, \mathbf{y})$ denoting the Bregman divergence. However, the global constant $L$ is still required to tune the algorithm, and studying this notion is beyond the scope of our paper.

### A.3 Lipschitz-Adaptive Algorithms

The upper bound of gradients $G$ and the diameter of the bounded feasible domain $D$ are often required to build up online algorithms. An algorithm that requires the diameter $D$ of the bounded feasible domain $D$ but not the upper bound of gradients is known to be Lipschitz-adaptive [Duchi et al., 2011; Orabona and Pál, 2018; Cutkosky, 2019; Mhammedi et al., 2019]. For the general OCO setting, to the best of our knowledge, there are *no* Lipschitz-adaptive algorithm that ensures a gradient-variation bound. When specialized to the Prediction with Experts' Advice (PEA) setting [Cesa-Bianchi and Lugosi, 2006], Chen et al. [2021] design a two-layer algorithm with a restarting mechanism that can obtain a gradient-variation bound in PEA setting. In this paper, we develop a new Lipschitz-adaptive algorithm with a lightweight design, which guarantees the gradient-variation bound as well and is the only option for our purposes, to the best of our knowledge.

## B  Omitted Details for Section 2

In this section, we first provide a judgement of the necessity of smoothness for first-order online algorithms in Appendix B.1. Next, we will provide proofs for the theorems in Section 2.3. In Appendix B.2, we present the omitted discussion for the challenge in designing the algorithm for exp-concave functions. We introduce a key lemma in Appendix B.3, which abstracts the key idea of exploiting the generalized smoothness. Later, the proofs for Theorem 1 and Theorem 2 are provided in Appendix B.4 and Appendix B.5, respectively.

### B.1  Necessity of Smoothness

We first provide a lower bound on the convergence rate for first-order methods with convex functions.

**Proposition 1** (Theorem 3.2.1 of Nesterov [2018]). *There exists a $G$-Lipschitz and convex function $f : \mathbb{R}^d \mapsto \mathbb{R}$ with $\|\mathbf{x}_1 - \mathbf{x}_\star\| \leq D$ where $\mathbf{x}_\star \in \arg\min_{\mathbf{x} \in \mathbb{R}^d} f(\mathbf{x})$ such that*

$$f(\mathbf{x}_t) - \min_{\mathbf{x} \in \mathbb{R}^d} f(\mathbf{x}) \geq \frac{GD}{2(2 + \sqrt{t})},$$

*for any optimization scheme that generates a sequence $\{\mathbf{x}_t\}$ satisfying that*

$$\mathbf{x}_t \in \mathbf{x}_1 + \text{Lin}\left\{\nabla f(\mathbf{x}_1), \dots, \nabla f(\mathbf{x}_{t-1})\right\},$$

*where $\text{Lin}\{\mathbf{a}_1, \dots, \mathbf{a}_t\}$ denotes the linear span of vectors $\mathbf{a}_1, \dots, \mathbf{a}_t$.*

Yang et al. [2014] prove the necessity of the smoothness to obtain the gradient-variation bounds with convex functions using the first-order online algorithms. We formally present this idea in below.

**Proposition 2** (Remark 1 of Yang et al. [2014]). *The smoothness assumption for $G$-Lipschitz and convex online functions $f_t : \mathcal{X} \mapsto \mathbb{R}$ is necessary for any online algorithms, whose decisions are linear combinations of the queried gradients when no projections are made, and ensure:*

$$\sum_{t=1}^T f_t(\mathbf{x}_t) - \min_{\mathbf{x} \in \mathcal{X}} \sum_{t=1}^T f_t(\mathbf{x}) \leq \mathcal{O}(\sqrt{V_T}) + constant,$$

*with only $c \cdot T$ times gradient queries, with $c \in \mathbb{N}$ being a constant independent of $T$ and environmental parameters, such as the Lipschitz constant and the smoothness constant.*

*Proof.* We prove this proposition by contradiction. Assume that $f_t$ is non-smooth. Then consider a special case where the algorithm projects the decisions onto $\mathcal{X} = \mathbb{R}^d$, i.e., no projections are made,

and all the online functions are equal $f_1 = \cdots = f_T = f$. Notice that, under such case, the gradient variation is zero and therefore,

$$\sum_{t=1}^{T} f(\mathbf{x}_t) - \min_{\mathbf{x} \in \mathcal{X}} \sum_{t=1}^{T} f(\mathbf{x}) \le \mathcal{O}(1),$$

which implies $\bar{\mathbf{x}}_T = \frac{1}{T} \sum_{t=1}^{T} \mathbf{x}_t$ approaches an $\mathcal{O}(1/T)$ convergence rate. Now it remains to check whether this convergence rate contradicts with Proposition 1. Denoted by $\mathbf{x}'_1, \ldots, \mathbf{x}'_{cT}$ the points that the algorithm queries gradients on, because the decisions are linear combinations of the gradients when no projections are made, then,

$$\bar{\mathbf{x}}_T \in \mathbf{x}_1 + \text{Lin} \{\nabla f(\mathbf{x}'_1), \ldots, \nabla f(\mathbf{x}'_{cT})\},$$

which indicates that, there exists a method which promises $\mathcal{O}(c/T) = \mathcal{O}(1/T)$ convergence rate under the protocol considered in Proposition 1, contradicting the lower bound. $\qquad\square$

## B.2 Challenge for Exp-Concave Functions in Regret Minimization

In Section 2, we design algorithms for convex and strongly convex functions respectively. However, the gradient-variation regret for exp-concave functions [Hazan et al., 2007] under generalized smoothness has not yet been addressed. For online learning with global smooth functions, it has been demonstrated that an $\mathcal{O}(d \log V_T)$ regret is attainable for exp-concave functions [Chiang et al., 2012], which is realized by an optimistic variant of the online Newton step algorithm [Hazan et al., 2007] using the last-round gradient as optimism, i.e., $M_t = \nabla f_{t-1}(\mathbf{x}_{t-1})$. However, it remains unclear how to obtain a general optimistic bound of order $\mathcal{O}(d \log D_T)$ with $D_T = \sum_{t=1}^{T} \|\nabla f_t(\mathbf{x}_t) - M_t\|_2^2$ for arbitrary optimism $\{M_t\}_{t=1}^{T}$. Our technique for achieving gradient-variation regret under generalized smoothness relies on a flexible setting for optimism (which may not be the last-round gradient) and step size tuning (which requires a trajectory-wise stability analysis), making it challenging for extension to exp-concave functions. We leave this as an open question for future research.

## B.3 Lemma for Regret Minimization

Below, we present a lemma that leverages the local smoothness of the optimization trajectory to derive gradient variation within the OMD framework. This lemma can be applied in the analysis of both convex and strongly convex functions.

**Lemma 2.** *Under Assumptions 1 and 2, by selecting regularizer as $\psi_t(\mathbf{x}) = \frac{1}{2\eta_t}\|\mathbf{x}\|_2^2$, setting step sizes satisfying that $\eta_{t+1} \le \eta_t$ and $\eta_t \le 1/(4\widehat{L}_{t-1})$, where $\widehat{L}_{t-1} = \ell_{t-1}(2\|\nabla f_{t-1}(\widehat{\mathbf{x}}_t)\|_2)$, and by choosing optimism as $M_t = \nabla f_{t-1}(\widehat{\mathbf{x}}_t)$, the OMD in Eq. (4) ensures the regret bound:*

$$\sum_{t=1}^{T} \langle \nabla f_t(\mathbf{x}_t), \mathbf{x}_t - \mathbf{x}_\star \rangle \le \sum_{t=1}^{T} \frac{1}{2\eta_t} \left( \|\mathbf{x}_\star - \widehat{\mathbf{x}}_t\|_2^2 - \|\mathbf{x}_\star - \widehat{\mathbf{x}}_{t+1}\|_2^2 \right)$$

$$+ 2\sum_{t=1}^{T} \eta_t \|\nabla f_t(\mathbf{x}_t) - \nabla f_{t-1}(\mathbf{x}_t)\|_2^2 - \sum_{t=1}^{T} \frac{1}{4\eta_t} \|\mathbf{x}_t - \widehat{\mathbf{x}}_t\|_2^2$$

$$\le \frac{D^2}{2\eta_T} + 2\sum_{t=1}^{T} \eta_t \|\nabla f_t(\mathbf{x}_t) - \nabla f_{t-1}(\mathbf{x}_t)\|_2^2 - \sum_{t=1}^{T} \frac{1}{4\eta_t} \|\mathbf{x}_t - \widehat{\mathbf{x}}_t\|_2^2, \quad (11)$$

*where $\mathbf{x}_\star \in \arg\min_{\mathbf{x} \in \mathcal{X}} \sum_{t=1}^{T} f_t(\mathbf{x})$ denotes the best decision in hindsight.*

*Proof.* Following the standard analysis of OMD (Lemma 3), OMD with the optimism chosen as $M_t = \nabla f_{t-1}(\widehat{\mathbf{x}}_t)$ ensures the following regret bound:

$$\sum_{t=1}^{T} \langle \nabla f_t(\mathbf{x}_t), \mathbf{x}_t - \mathbf{x}_\star \rangle \le \underbrace{\sum_{t=1}^{T} \frac{1}{2\eta_t} \left( \|\mathbf{x}_\star - \widehat{\mathbf{x}}_t\|_2^2 - \|\mathbf{x}_\star - \widehat{\mathbf{x}}_{t+1}\|_2^2 \right)}_{\text{TERM-A}} + \underbrace{\sum_{t=1}^{T} \eta_t \|\nabla f_t(\mathbf{x}_t) - \nabla f_{t-1}(\widehat{\mathbf{x}}_t)\|_2^2}_{\text{TERM-B}}$$

$$-\sum_{t=1}^{T}\frac{1}{2\eta_t}\left(\|\widehat{\mathbf{x}}_{t+1}-\mathbf{x}_t\|_2^2+\|\mathbf{x}_t-\widehat{\mathbf{x}}_t\|_2^2\right).$$

$$\underbrace{\phantom{-\sum_{t=1}^{T}\frac{1}{2\eta_t}\left(\|\widehat{\mathbf{x}}_{t+1}-\mathbf{x}_t\|_2^2+\|\mathbf{x}_t-\widehat{\mathbf{x}}_t\|_2^2\right).}}_{\text{TERM-C}}$$

The analysis for TERM-A is straightforward under Assumption 2:

$$\text{TERM-A}\le\frac{1}{2\eta_1}\|\mathbf{x}_\star-\widehat{\mathbf{x}}_1\|_2^2+\sum_{t=2}^{T}(\frac{1}{2\eta_t}-\frac{1}{2\eta_{t-1}})\|\mathbf{x}_\star-\widehat{\mathbf{x}}_t\|_2^2\le\frac{D^2}{2\eta_1}+\sum_{t=2}^{T}(\frac{D^2}{2\eta_t}-\frac{D^2}{2\eta_{t-1}})=\frac{D^2}{2\eta_T}.$$

Next, we analyze the TERM-B under the generalized smoothness condition. By the basic calculation, we can decompose the TERM-B into following two terms:

$$\text{TERM-B}\le 2\sum_{t=1}^{T}\eta_t\|\nabla f_t(\mathbf{x}_t)-\nabla f_{t-1}(\mathbf{x}_t)\|_2^2+2\sum_{t=2}^{T}\eta_t\|\nabla f_{t-1}(\mathbf{x}_t)-\nabla f_{t-1}(\widehat{\mathbf{x}}_t)\|_2^2,\quad(12)$$

where the first term is the desirable gradient variation and the second term should be further analyzed under the generalized smoothness. Given the optimism setting and the step size tuning, we demonstrate that $\mathbf{x}_t$ and $\widehat{\mathbf{x}}_t$ are sufficiently close to apply local smoothness:

$$\|\mathbf{x}_t-\widehat{\mathbf{x}}_t\|_2\le\eta_t\|\nabla f_{t-1}(\widehat{\mathbf{x}}_t)\|_2\le\frac{\|\nabla f_{t-1}(\widehat{\mathbf{x}}_t)\|_2}{4\widehat{L}_{t-1}}.$$

In above, the first inequality is by the the Pythagorean theorem and the second inequality is by the step size tuning. Therefore, we can apply Lemma 1 to bound the gradient deviation in Eq. (12) by

$$\sum_{t=2}^{T}\eta_t\|\nabla f_{t-1}(\mathbf{x}_t)-\nabla f_{t-1}(\widehat{\mathbf{x}}_t)\|_2\le\sum_{t=2}^{T}\eta_t\widehat{L}_{t-1}^2\|\mathbf{x}_t-\widehat{\mathbf{x}}_t\|_2^2.$$

Finally, combining TERM-A, TERM-B and TERM-C together, we obtain:

$$\sum_{t=1}^{T}\langle\nabla f_t(\mathbf{x}_t),\mathbf{x}_t-\mathbf{x}_\star\rangle\le\frac{D^2}{2\eta_T}+2\sum_{t=1}^{T}\eta_t\|\nabla f_t(\mathbf{x}_t)-\nabla f_{t-1}(\mathbf{x}_t)\|_2^2-\sum_{t=1}^{T}\frac{1}{4\eta_t}\|\mathbf{x}_t-\widehat{\mathbf{x}}_t\|_2^2$$

$$+\sum_{t=1}^{T}(2\eta_t\widehat{L}_{t-1}^2-\frac{1}{4\eta_t})\|\mathbf{x}_t-\widehat{\mathbf{x}}_t\|_2^2$$

$$\le\frac{D^2}{2\eta_T}+2\sum_{t=1}^{T}\eta_t\|\nabla f_t(\mathbf{x}_t)-\nabla f_{t-1}(\mathbf{x}_t)\|_2^2-\sum_{t=1}^{T}\frac{1}{4\eta_t}\|\mathbf{x}_t-\widehat{\mathbf{x}}_t\|_2^2,$$

where the second inequality is by setting $\eta_t\le 1/(4\widehat{L}_{t-1})$. Hence, we finish the proof. $\qquad\square$

### B.4 Proof of Theorem 1

*Proof.* The step size tuning in Eq. (7) in Theorem 1 satisfies the criterions for applying Lemma 2, specifically that $\eta_{t+1}\le\eta_t$ and $\eta_t\le 1/(4\widehat{L}t-1)$, where $\widehat{L}_{t-1}=\ell_{t-1}(2\|\nabla f_{t-1}(\widehat{\mathbf{x}}_t)\|_2)$. Therefore, by Lemma 2 and the convexity of loss functions, we obtain:

$$\sum_{t=1}^{T}f_t(\mathbf{x}_t)-\sum_{t=1}^{T}f_t(\mathbf{x}_\star)\le\sum_{t=1}^{T}\langle\nabla f_t(\mathbf{x}_t),\mathbf{x}_t-\mathbf{x}_\star\rangle\le\frac{D^2}{2\eta_T}+2\sum_{t=1}^{T}\eta_t\|\nabla f_t(\mathbf{x}_t)-\nabla f_{t-1}(\mathbf{x}_t)\|_2^2,$$

where $\mathbf{x}_\star\in\arg\min_{\mathbf{x}\in\mathcal{X}}\sum_{t=1}^{T}f_t(\mathbf{x})$. The first term can be bounded as follows,

$$\frac{D^2}{2\eta_T}\le 2\widehat{L}_{\max}D^2+\frac{D}{2}\sqrt{1+\sum_{t=2}^{T}\|\nabla f_t(\mathbf{x}_t)-\nabla f_{t-1}(\mathbf{x}_t)\|_2^2}+\frac{D}{2}\cdot\|\nabla f_1(\mathbf{x}_1)\|_2$$

$$=\mathcal{O}\left(\widehat{L}_{\max}D^2+D\sqrt{V_T}\right),$$

where we denote by $\widehat{L}_{\max} = \max_{t\in[T]} \widehat{L}_t$. For the second term, we apply the self-confident tuning lemma (Lemma 5) by choosing $f(x) = 1/\sqrt{x}$ in the lemma statement:

$$\sum_{t=1}^{T} \eta_t \|\nabla f_t(\mathbf{x}_t) - \nabla f_{t-1}(\mathbf{x}_t)\|_2^2 \leq \sum_{t=1}^{T} \frac{D\|\nabla f_t(\mathbf{x}_t) - \nabla f_{t-1}(\mathbf{x}_t)\|_2^2}{\sqrt{1 + \sum_{s=1}^{t-1} \|\nabla f_s(\mathbf{x}_s) - \nabla f_{s-1}(\mathbf{x}_s)\|_2^2}}$$

$$\leq 2D\sqrt{1 + \sum_{s=1}^{T} \|\nabla f_s(\mathbf{x}_s) - \nabla f_{s-1}(\mathbf{x}_s)\|_2^2} + D\max_{t\in[T]}\|\nabla f_t(\mathbf{x}_t) - \nabla f_{t-1}(\mathbf{x}_t)\|_2^2$$

$$= \mathcal{O}\left(D\sqrt{V_T} + \widehat{G}_{\max}^2 D\right),$$

where we use $\widehat{G}_{\max}$ to denote the empirically maximum Lipschitz constant. The additional term $\mathcal{O}(\widehat{G}_{\max}^2 D)$ results from the lack of knowledge about $\widehat{G}_{\max}$. However, we can improve this term to $\mathcal{O}(\widehat{G}_{\max} D)$ by incorporating the clipping technique [Cutkosky, 2019; Chen et al., 2021] into OMD framework. In the main text, we avoid introducing this method to prevent complicating the approach further. Our goal in the OMD introduction is to illustrate how to adapt to generalized smoothness at the base level. The details of this refined approach are provided in Appendix B.6. $\qquad\square$

## B.5   Proof of Theorem 2

*Proof.* For $\lambda$-strongly convex functions, we tune the step size as $\eta_t = 2/(\lambda t + 16\max_{s\in[t-1]}\widehat{L}_s)$, where we denote by $\widehat{L}_t = \ell_t(2\|\nabla f_t(\widehat{\mathbf{x}}_{t+1})\|)$ the locally estimated smoothness constant. By the property of strongly convex functions and Eq. (11) in Lemma 2, we have:

$$\sum_{t=1}^{T} f_t(\mathbf{x}_t) - \sum_{t=1}^{T} f_t(\mathbf{x}_\star) \leq \sum_{t=1}^{T} \langle \nabla f_t(\mathbf{x}_t), \mathbf{x}_t - \mathbf{x}_\star \rangle - \frac{\lambda}{2}\|\mathbf{x}_\star - \mathbf{x}_t\|_2^2$$

$$\leq \underbrace{\sum_{t=1}^{T} \frac{1}{2\eta_t}\left(\|\mathbf{x}_\star - \widehat{\mathbf{x}}_t\|_2^2 - \|\mathbf{x}_\star - \widehat{\mathbf{x}}_{t+1}\|_2^2\right) - \frac{\lambda}{2}\|\mathbf{x}_\star - \mathbf{x}_t\|_2^2}_{\text{TERM-A}} + \underbrace{2\sum_{t=1}^{T} \eta_t\|\nabla f_t(\mathbf{x}_t) - \nabla f_{t-1}(\mathbf{x}_t)\|_2^2}_{\text{TERM-B}}$$

$$\underbrace{-\sum_{t=1}^{T} \frac{1}{4\eta_t}\|\mathbf{x}_t - \widehat{\mathbf{x}}_t\|_2^2}_{\text{TERM-C}}.$$

Unlike in the case of convex functions, TERM-A involves negative terms derived from the strong convexity of the loss functions, which require a slightly different analysis:

$$\text{TERM-A} \leq \frac{1}{2\eta_1}\|\mathbf{x}_\star - \widehat{\mathbf{x}}_1\|_2^2 + \sum_{t=2}^{T}\left(\frac{1}{2\eta_t} - \frac{1}{2\eta_{t-1}}\right)\|\mathbf{x}_\star - \widehat{\mathbf{x}}_t\|_2^2 - \frac{\lambda}{2}\|\mathbf{x}_\star - \mathbf{x}_t\|_2^2$$

$$\leq \frac{\lambda D^2}{4} + \sum_{t=2}^{T}\left(\frac{\lambda}{4} + 4\max_{s\in[t]}\widehat{L}_s - 4\max_{s\in[t-1]}\widehat{L}_s\right)\|\mathbf{x}_\star - \widehat{\mathbf{x}}_t\|_2^2 - \frac{\lambda}{2}\|\mathbf{x}_\star - \mathbf{x}_t\|_2^2$$

$$\leq \frac{\lambda D^2}{2} + 4D^2 \cdot \max_{s\in[T]}\widehat{L}_s + \sum_{t=2}^{T}\frac{\lambda}{4}\|\mathbf{x}_\star - \widehat{\mathbf{x}}_t\|_2^2 - \frac{\lambda}{2}\|\mathbf{x}_\star - \mathbf{x}_t\|_2^2 \quad \text{(bounded domain assumption)}$$

$$\leq \frac{\lambda D^2}{4} + 4D^2\widehat{L}_{\max} + \sum_{t=2}^{T}\frac{\lambda}{2}\|\mathbf{x}_t - \widehat{\mathbf{x}}_t\|_2^2 \quad (\|\mathbf{x}_\star - \widehat{\mathbf{x}}_t\|_2^2 \leq 2\|\mathbf{x}_\star - \mathbf{x}_t\|_2^2 + 2\|\mathbf{x}_t - \widehat{\mathbf{x}}_t\|_2^2)$$

$$\leq \frac{\lambda D^2}{4} + 4D^2\widehat{L}_{\max} + \sum_{t=2}^{T}\frac{\lambda \cdot \eta_t^2}{2}\|\nabla f_t(\mathbf{x}_t) - \nabla f_{t-1}(\widehat{\mathbf{x}}_t)\|_2^2 \quad \text{(Lemma 4)}$$

$$\leq \frac{\lambda D^2}{4} + 4D^2\widehat{L}_{\max} + \sum_{t=1}^{T}\eta_t\|\nabla f_t(\mathbf{x}_t) - \nabla f_{t-1}(\widehat{\mathbf{x}}_t)\|_2^2$$

$$\leq \frac{\lambda D^2}{4} + 4D^2 \widehat{L}_{\max} + \sum_{t=1}^{T} 2\eta_t \|\nabla f_t(\mathbf{x}_t) - \nabla f_{t-1}(\mathbf{x}_t)\|_2^2 + \sum_{t=1}^{T} 2\eta_t \|\nabla f_{t-1}(\mathbf{x}_t) - \nabla f_{t-1}(\widehat{\mathbf{x}}_t)\|_2^2,$$

where the last term can be cancelled by TERM-C, and the third term is in the same order of TERM-B. Therefore, we only need to focus on TERM-B. For TERM-B, we follow the analysis of Chen et al. [2024], who applies a simpler analysis for the self-confident tuning. First, we define:

$$\alpha = \left\lceil \sum_{t=1}^{T} \|\nabla f_t(\mathbf{x}_t) - \nabla f_{t-1}(\mathbf{x}_t)\|_2^2 \right\rceil.$$

Then by dividing the time horizon into $[1, \alpha]$ and $[\alpha + 1, T]$, we can upper bound TERM-B as:

$$
\begin{aligned}
\text{TERM-B} &\leq 2 \sum_{t=1}^{\alpha} \eta_t \|\nabla f_t(\mathbf{x}_t) - \nabla f_{t-1}(\mathbf{x}_t)\|_2^2 + 2 \sum_{t=\alpha+1}^{T} \eta_t \|\nabla f_t(\mathbf{x}_t) - \nabla f_{t-1}(\mathbf{x}_t)\|_2^2 \\
&\leq 4 \sum_{t=1}^{\alpha} \frac{1}{\lambda t} \|\nabla f_t(\mathbf{x}_t) - \nabla f_{t-1}(\mathbf{x}_t)\|_2^2 + 4 \sum_{t=\alpha+1}^{T} \frac{1}{\lambda t} \|\nabla f_t(\mathbf{x}_t) - \nabla f_{t-1}(\mathbf{x}_t)\|_2^2 \\
&\leq 4 \left( \max_{s \in [T]} \|\nabla f_s(\mathbf{x}_s) - \nabla f_{s-1}(\mathbf{x}_s)\|_2^2 \right) \sum_{t=1}^{\alpha} \frac{1}{\lambda t} + 4 \sum_{t=\alpha+1}^{T} \frac{1}{\lambda t} \|\nabla f_t(\mathbf{x}_t) - \nabla f_{t-1}(\mathbf{x}_t)\|_2^2 \qquad (13) \\
&\leq 16 \widehat{G}_{\max}^2 \sum_{t=1}^{\alpha} \frac{1}{\lambda t} + \frac{4}{\lambda(\alpha+1)} \sum_{t=\alpha+1}^{T} \|\nabla f_t(\mathbf{x}_t) - \nabla f_{t-1}(\mathbf{x}_t)\|_2^2 \\
&\leq \frac{16 \widehat{G}_{\max}^2}{\lambda} (1 + \ln \alpha) + \frac{4}{\lambda} \\
&\leq \frac{16 \widehat{G}_{\max}^2}{\lambda} \ln \left( \sum_{t=1}^{T} \|\nabla f_t(\mathbf{x}_t) - \nabla f_{t-1}(\mathbf{x}_t)\|_2^2 + 1 \right) + \frac{16 \widehat{G}_{\max}^2 + 4}{\lambda},
\end{aligned}
$$

where we use $\widehat{G}_{\max}$ to denote the maximum Lipschitz constant estimated empirically. Therefore:

$$\sum_{t=1}^{T} f_t(\mathbf{x}_t) - \sum_{t=1}^{T} f_t(\mathbf{x}_\star) \leq \mathcal{O}\left( \frac{\widehat{G}_{\max}^2}{\lambda} \log V_T + \widehat{L}_{\max} D^2 + \lambda D^2 \right),$$

which finishes the proof. $\qquad \square$

## B.6 OMD Incorporating Clipping Technique

In Appendix B.4, an additional term $\mathcal{O}(\widehat{G}_{\max}^2 D)$ shows up in the final regret bound, which results from the lack of knowledge about $\widehat{G}_{\max}$. In this subsection, we improve the term $\mathcal{O}(\widehat{G}_{\max}^2 D)$ to $\mathcal{O}(\widehat{G}_{\max} D)$ by incorporating the clipping technique [Cutkosky, 2019; Chen et al., 2021] into the OMD framework. This modified OMD algorithm is defined as follows,

$$\mathbf{x}_t = \Pi_{\mathcal{X}} \left[ \widehat{\mathbf{x}}_t - \eta_t \nabla f_{t-1}(\widehat{\mathbf{x}}_t) \right], \quad \widehat{\mathbf{x}}_{t+1} = \Pi_{\mathcal{X}} \left[ \widehat{\mathbf{x}}_t - \eta_t \widetilde{\mathbf{g}}_t \right], \qquad (14)$$

where $\widetilde{\mathbf{g}}_t = \nabla f_{t-1}(\widehat{\mathbf{x}}_t) + \frac{B_{t-1}}{B_t} (\nabla f_t(\mathbf{x}_t) - \nabla f_{t-1}(\widehat{\mathbf{x}}_t))$ is a clipped gradient with the maintained threshold $B_t = \max\{B_0, \max_{s \in [t]} \|\nabla f_s(\mathbf{x}_s) - \nabla f_{s-1}(\widehat{\mathbf{x}}_s)\|_2\}$. Notice that, $\mathbf{x}_t$ still updates from $\widehat{\mathbf{x}}_t$ in the same manner as illustrated in Theorem 1, therefore, we can apply the similar analysis to it when exploiting the smoothness locally. Correspondingly, we provided the following step size tuning:

$$\eta_t = \min \left\{ \sqrt{\frac{D^2}{B_{t-1}^2 + \sum_{s=1}^{t-1} \|\widetilde{\mathbf{g}}_s - \nabla f_{s-1}(\widehat{\mathbf{x}}_s)\|_2^2}}, \ \min_{s \in [t]} \frac{1}{4 \widehat{L}_{s-1}} \right\}, \qquad (15)$$

where the key modification is that we add $B_{t-1}^2$ in the denominator to facilitate the tuning analysis and we denote by $\widehat{L}_t = \ell_t(2\|\nabla f_t(\widehat{\mathbf{x}}_{t+1})\|)$. The following theorem presents the guarantee.

**Theorem 5.** *Under Assumptions 1 - 2, assuming online functions are convex, OMD presented in Eq.* (14) *with the step sizes in Eq.* (15) *ensures that:*

$$\mathrm{REG}_T \leq \frac{5D}{2}\sqrt{2V_T} + 4\widehat{L}_{\max}D^2 + 5\widehat{G}_{\max}D,$$

*where $V_T = \sum_{t=2}^{T} \sup_{\mathbf{x} \in \mathcal{X}} \|\nabla f_t(\mathbf{x}) - \nabla f_{t-1}(\mathbf{x})\|_2^2$ is the gradient variations, $\widehat{L}_{\max} = \max_{t \in [T]} \widehat{L}_t$ is the maximum smoothness constant over the optimization trajectory, and $\widehat{G}_{\max}$ is the maximum empirically estimated Lipschitz constant.*

*Proof.* First, we prove that the clipping technique incurs a constant in the regret bound:

$$\sum_{t=1}^{T}\langle\nabla f_t(\mathbf{x}_t), \mathbf{x}_t - \mathbf{x}_\star\rangle - \langle\widetilde{\mathbf{g}}_t, \mathbf{x}_t - \mathbf{x}_\star\rangle = \sum_{t=1}^{T}\frac{B_t - B_{t-1}}{B_t}\langle\nabla f_t(\mathbf{x}_t) - \nabla f_{t-1}(\widehat{\mathbf{x}}_t), \mathbf{x}_t - \mathbf{x}_\star\rangle$$

$$\leq D\sum_{t=1}^{T}(B_t - B_{t-1}) = D(B_T - B_0) = \mathcal{O}(\widehat{G}_{\max}D).$$

In the following analysis, we will focus on the regret associated with the clipped gradient $\widetilde{\mathbf{g}}_t$. Following the standard analysis of OMD (Lemma 3), and with simple calculations, we arrive at:

$$\sum_{t=1}^{T}\langle\widetilde{\mathbf{g}}_t, \mathbf{x}_t - \mathbf{x}_\star\rangle \leq \underbrace{\frac{D^2}{2\eta_T}}_{\text{TERM-A}} + \underbrace{\sum_{t=1}^{T}\eta_t\|\widetilde{\mathbf{g}}_t - \nabla f_{t-1}(\widehat{\mathbf{x}}_t)\|_2^2}_{\text{TERM-B}} - \underbrace{\sum_{t=1}^{T}\frac{1}{2\eta_t}\|\mathbf{x}_t - \widehat{\mathbf{x}}_t\|_2^2}_{\text{TERM-C}}. \quad (16)$$

By the step size tuning, TERM-A can be bounded as:

$$\mathrm{TERM\text{-}A} \leq 2\widehat{L}_{\max}D^2 + \frac{D}{2}\sqrt{B_T^2 + \sum_{s=1}^{T}\|\widetilde{\mathbf{g}}_s - \nabla f_{s-1}(\widehat{\mathbf{x}}_s)\|_2^2}$$

$$\leq 2\widehat{L}_{\max}D^2 + \widehat{G}_{\max}D + \frac{D}{2}\sqrt{\sum_{s=2}^{T}\|\widetilde{\mathbf{g}}_s - \nabla f_{s-1}(\widehat{\mathbf{x}}_s)\|_2^2}$$

$$= 2\widehat{L}_{\max}D^2 + \widehat{G}_{\max}D + \frac{D}{2}\sqrt{\sum_{s=2}^{T}\frac{B_{s-1}^2}{B_s^2}\|\nabla f_s(\mathbf{x}_s) - \nabla f_{s-1}(\widehat{\mathbf{x}}_s)\|_2^2}$$

$$\leq 2\widehat{L}_{\max}D^2 + \widehat{G}_{\max}D + \underbrace{\frac{D}{2}\sqrt{\sum_{s=2}^{T}\|\nabla f_s(\mathbf{x}_s) - \nabla f_{s-1}(\widehat{\mathbf{x}}_s)\|_2^2}}_{\text{TERM-A-VAR}}. \quad (17)$$

For TERM-B in (16), by the self-confident tuning lemma (Lemma 6) we obtain,

$$\mathrm{TERM\text{-}B} = D\sum_{t=1}^{T}\frac{\|\widetilde{\mathbf{g}}_t - \nabla f_{t-1}(\widehat{\mathbf{x}}_t)\|_2^2}{\sqrt{B_{t-1}^2 + \sum_{s=1}^{t-1}\|\widetilde{\mathbf{g}}_s - \nabla f_{s-1}(\widehat{\mathbf{x}}_s)\|_2^2}} \leq D\sum_{t=1}^{T}\frac{\|\widetilde{\mathbf{g}}_t - \nabla f_{t-1}(\widehat{\mathbf{x}}_t)\|_2^2}{\sqrt{\sum_{s=1}^{t}\|\widetilde{\mathbf{g}}_s - \nabla f_{s-1}(\widehat{\mathbf{x}}_s)\|_2^2}}$$

$$\leq 2D\sqrt{\sum_{t=1}^{T}\|\widetilde{\mathbf{g}}_s - \nabla f_{t-1}(\widehat{\mathbf{x}}_t)\|_2^2} \leq \underbrace{2D\sqrt{\sum_{t=2}^{T}\|\nabla f_t(\mathbf{x}_t) - \nabla f_{t-1}(\widehat{\mathbf{x}}_t)\|_2^2}}_{\text{TERM-B-VAR}} + 2\widehat{G}_{\max}D. \quad (18)$$

Combine TERM-A-VAR in (17), TERM-B-VAR in (18), and negative term TERM-C in (16):

TERM-A-VAR + TERM-B-VAR − TERM-C

$$= \frac{5D}{2}\sqrt{2\sum_{t=2}^{T}\|\nabla f_t(\mathbf{x}_t) - \nabla f_{t-1}(\mathbf{x}_t)\|_2^2 + 2\sum_{t=2}^{T}\|\nabla f_{t-1}(\mathbf{x}_t) - \nabla f_{t-1}(\widehat{\mathbf{x}}_t)\|_2^2} - \sum_{t=1}^{T}\frac{1}{2\eta_t}\|\mathbf{x}_t - \widehat{\mathbf{x}}_t\|_2^2$$

$$\leq \frac{5D}{2}\sqrt{2\sum_{t=2}^{T}\|\nabla f_t(\mathbf{x}_t) - \nabla f_{t-1}(\mathbf{x}_t)\|_2^2} + \frac{5D}{2}\sqrt{2\sum_{t=2}^{T}\|\nabla f_{t-1}(\mathbf{x}_t) - \nabla f_{t-1}(\widehat{\mathbf{x}}_t)\|_2^2}$$

$$\quad - \sum_{t=1}^{T}\frac{1}{2\eta_t}\|\mathbf{x}_t - \widehat{\mathbf{x}}_t\|_2^2$$

$$\leq \frac{5D}{2}\sqrt{2V_T} + \frac{5D}{2}\sqrt{2\sum_{t=2}^{T}\widehat{L}_{t-1}^2\|\mathbf{x}_t - \widehat{\mathbf{x}}_t\|_2^2} - \sum_{t=2}^{T}2\big(\max_{s\in[t-1]}\widehat{L}_s\big)\|\mathbf{x}_t - \widehat{\mathbf{x}}_t\|_2^2$$

$$\leq \frac{5D}{2}\sqrt{2V_T} + \frac{25}{16}\widehat{L}_{\max}D^2,$$

where in the forth line we apply Lemma 1, and ub the final line, we apply the AM-GM inequality, $2\sqrt{ab} - a \leq b$. Combing each component together, we conclude the proof as:

$$\sum_{t=1}^{T}f_t(\mathbf{x}) - \sum_{t=1}^{T}f_t(\mathbf{x}_\star) \leq \sum_{t=1}^{T}\langle\nabla f_t(\mathbf{x}_t), \mathbf{x}_t - \mathbf{x}_\star\rangle \leq \sum_{t=1}^{T}\langle\widetilde{\mathbf{g}}_t, \mathbf{x}_t - \mathbf{x}_\star\rangle + 2\widehat{G}_{\max}D$$

$$\leq \frac{5D}{2}\sqrt{2V_T} + 4\widehat{L}_{\max}D^2 + 5\widehat{G}_{\max}D.$$

$$\square$$

## C  Omitted Details for Section 3

In this section, we present the omitted details for Section 3. Appendix C.1 discusses how set the optimism $\boldsymbol{m}_t$ efficiently via a binary search. Appendix C.2 provides the omitted discussion for the challenge in universal online learning with exp-concave functions. Proofs for the theorems in Section 3 are included in Appendix C.3 and Appendix C.4. Appendix C.5 gives a simple universal algorithm under the global smoothness condition, which improves the optimality and efficiency of the method by Yan et al. [2023], at a cost of additional function value queries.

### C.1  Discussion on Optimism

In this part, we discuss how to set the optimism $\boldsymbol{m}_t$ that involves the decision $\boldsymbol{p}_t$. We present the steps for incorporating the optimism and generating the decision $\boldsymbol{p}_t$ in Algorithm 1 for reference:

$$\widetilde{w}_{t,i} = w_{t,i}\exp(\eta_{t,i}m_{t,i}), \quad p_{t,i} = \frac{\eta_{t,i}\widetilde{w}_{t,i}}{\sum_{j\in[N]}\eta_{t,j}\widetilde{w}_{t,j}}. \tag{19}$$

For more general consideration, we set optimism the as $m_{t,i} = f(\sum_{i\in[N]}p_{t,i}\mathbf{x}_{t,i}) - h(\mathbf{x}_{t,i})$ where $f : \mathcal{X} \mapsto \mathbb{R}$ is a convex and continuous function and $\mathbf{x}_i \in \mathcal{X}$ is a decision available before setting the optimism. Notice that $\widetilde{w}_i$ requires $\boldsymbol{p}_t$ to update while $\boldsymbol{p}_t$ is produced based on $\widetilde{w}_{t,i}$, resulting in a circular argument. Following Wei et al. [2016], the $\boldsymbol{p}_t$ can be solved via the binary search technique. We define $\alpha = f(\sum_{i\in[N]}p_{t,i}\mathbf{x}_i)$, then the weight $\widetilde{w}_{t,i}$ is a function of $\alpha$ as $\widetilde{w}_{t,i}(\alpha) = w_{t,i}\exp(\eta_{t,i}(\alpha - f(\mathbf{x}_{t,i})))$. Furthermore, with the update formulation in (19), the decision $p_{t,i}$ is a function of $\alpha$ as well, with the formulation $p_{t,i}(\alpha) = \frac{\eta_{t,i}\widetilde{w}_{t,i}(\alpha)}{\sum_{j\in[N]}\eta_{t,j}\widetilde{w}_{t,j}(\alpha)}$. By introducing a function $g(\alpha) = f(\sum_{i\in[N]}p_{t,i}(\alpha)\mathbf{x}_i)$, solving the decision $\boldsymbol{p}_t$ is equal to solving $g(\alpha) = \alpha$.

Below, we prove the existence of a solution to $g(\alpha) = \alpha$. Provided the lower bound $\underline{f}$ of function $f(\cdot)$, and by the convexity of the function $f(\cdot)$, the searching range of $\alpha$ is restricted to $\underline{f} \leq \alpha \leq \max_{i\in[N]}\{f(\mathbf{x}_i)\}$, and thus $\alpha$ is bounded. The continuity of the function $f(\cdot)$ implies the continuity of the function $g(\alpha)$ as well. The choice of $\alpha = \underline{f}$ results in $g(\alpha) - \alpha = f(\sum_{i\in[N]}p_{t,i}(\alpha)\mathbf{x}_i) - \underline{f} \geq 0$, and the choice of $\alpha = \max_{i\in[N]}\{f(\mathbf{x}_i)\}$ implies $g(\alpha) - \alpha \leq \sum_{i\in[N]}p_{t,i}(\alpha)f(\mathbf{x}_i) - \max_{i\in[N]}\{f(\mathbf{x}_i)\} \leq 0$, indicating that a solution to $g(\alpha) = \alpha$ exists. By using a binary search within $[\underline{f}, \max_{i\in[N]}\{f(\mathbf{x}_i)\}]$, we can approach $\alpha$ within an error $\mathcal{O}(1/T)$ in $\mathcal{O}(\log T)$ iterations.

The above argument requires the lower bound of the function $f(\cdot)$ to determine the searching range over $\alpha$. When considering a simpler case where $f(\mathbf{x}_i) = \ell_i$ and $f(\sum_{i \in [N]} p_{t,i} \mathbf{x}_i) = \sum_{i \in [N]} p_{t,i} \ell_i$, we can omit the requirement for the lower bound because $\alpha$ falls within $[\min_{i \in [N]}\{\ell_i\}, \max_{i \in [N]}\{\ell_i\}]$, because of the simpler structure of linear functions.

## C.2    Challenge for Exp-Concave Functions in Universal Online Learning

In Section 3.4, we present Algorithm 2 that achieves gradient-variation bounds for convex and strongly convex functions simultaneously. However, it does not guarantee such bounds for exp-concave functions. Besides the challenges discussed in Section B.2, a new obstacle arises in designing the meta-algorithm. We require optimism to satisfy $\langle \boldsymbol{p}_t, \boldsymbol{m}_t \rangle \leq 0$, since we pass the meta-algorithm with heterogeneous inputs, and therefore we set $m_{t,i} = f_{t-1}(\mathbf{x}_t) - f_{t-1}(\mathbf{x}_{t,i})$ for all the base-learners. This optimism design is suitable for strongly convex functions, as the term $\sqrt{\sum_t (f_{t-1}(\mathbf{x}_t) - f_{t-1}(\mathbf{x}_{t,i}))^2}$ introduced by optimism can be bounded by $\widehat{G}_{\max}\sqrt{\sum_t \|\mathbf{x}_t - \mathbf{x}_{t,i}\|_2^2}$, cancelled by the negative term $-\sum_t \lambda\|\mathbf{x}_t - \mathbf{x}_{t,i}\|_2^2$ from strong convexity. However, for exp-concave functions, the negative term $-\langle \nabla f_t(\mathbf{x}_t), \mathbf{x}_t - \mathbf{x}_{t,i} \rangle^2$ from exp-concavity may not be sufficient to cancel $\sqrt{\sum_t (f_{t-1}(\mathbf{x}_t) - f_{t-1}(\mathbf{x}_{t,i}))^2}$. We leave this as an open problem for future exploration.

## C.3    Proof of Theorem 3

In this subsection, we prove a slightly generalized version of Theorem 3, which does not specify optimism and instead imposes conditions only on $\bar{r}_t$. One can verify that the setting of optimism in Theorem 3 satisfies the requirement of Theorem 6.

**Theorem 6.** *By setting $\bar{r}_{t,i}$ such that $\sum_{i=1}^{N} p_{t,i}\bar{r}_{t,i} \leq 0$, Algorithm 1 ensures that, for any $i_\star \in [N]$, the regret $\sum_{t=1}^{T}\langle \boldsymbol{p}_t, \boldsymbol{\ell}_t \rangle - \sum_{t=1}^{T} \ell_{t,i_\star}$ can be bounded by:*

$$\mathcal{O}\left(\left(\sqrt{\sum_{t=1}^{T}(r_{t,i_\star} - m_{t,i_\star})^2} + B_T\right) \cdot \left(\log(N) + \log(B_T + \log T)\right)\right),$$

*where $B_T = \max\{B_0, \max_{t \in [T]}\|\boldsymbol{r}_t - \boldsymbol{m}_t\|_\infty\}$.*

*Proof.* First, we demonstrate that the clipping technique [Chen et al., 2021; Cutkosky, 2019] incurs a constant in the regret bound:

$$\sum_{t=1}^{T} r_{t,i_\star} - \bar{r}_{t,i_\star} = \sum_{t=1}^{T} r_{t,i_\star} - m_{t,i_\star} - \frac{B_{t-1}}{B_t}(r_{t,i_\star} - m_{t,i_\star}) = \sum_{t=1}^{T} \frac{B_t - B_{t-1}}{B_t}(r_{t,i_\star} - m_{t,i_\star})$$

$$\leq \sum_{t=1}^{T} \frac{B_t - B_{t-1}}{B_t}|r_{t,i_\star} - m_{t,i_\star}| \leq \sum_{t=1}^{T} \frac{B_t - B_{t-1}}{B_t}\|\boldsymbol{r}_t - \boldsymbol{m}_t\|_\infty \leq B_T - B_0. \tag{20}$$

In below, we focus on the analysis associated with clipped regret $\bar{r}_{t,i_\star}$. Following previous work [Wei et al., 2016], we define $W_t = \sum_{i=1}^{N} w_{t,i}$ to represent the summation of weights at time $t$. The quantity $W_t$ can be realized as the potential to be analyzed. Next, we consider to upper bound $\ln W_{T+1}$.

By the inequality $x \leq x^\alpha + (\alpha - 1)/e$ for $x > 0, \alpha \geq 0$, for any $i \in [N]$, we have:

$$w_{T+1,i} \leq (w_{T+1,i})^{\frac{\eta_{T,i}}{\eta_{T+1,i}}} + \frac{1}{e}\left(\frac{\eta_{T,i}}{\eta_{T+1,i}} - 1\right). \tag{21}$$

Based on the updates in Line 7 of Algorithm 1, we bound the first term on the right-hand side as:

$$(w_{T+1,i})^{\frac{\eta_{T,i}}{\eta_{T+1,i}}} = w_{T,i}\exp\left(\eta_{T,i}\bar{r}_{T,i} - \eta_{T,i}^2(\bar{r}_{T,i} - m_{T,i})^2\right)$$

$$= \widetilde{w}_{T,i}\exp\left(\eta_{T,i}(\bar{r}_{T,i} - m_{T,i}) - \eta_{T,i}^2(\bar{r}_{T,i} - m_{T,i})^2\right)$$

$$\leq \widetilde{w}_{T,i}\left(1 + \eta_{T,i}\left(\bar{r}_{T,i} - m_{T,i}\right)\right). \tag{22}$$

The last inequality is by $\exp(x - x^2) \leq 1 + x$ for $x \geq -1/2$. This is a crucial condition needed to be verified for Lipschitz-adaptive meta-algorithm. By the tuning of learning rates, and the clipping technique, we control the range of $\eta_{T,i}(\bar{r}_{T,i} - m_{T,i})$ well:

$$\eta_{T,i}|\bar{r}_{T,i} - m_{T,i}| = \eta_{T,i}\frac{B_{T-1}}{B_T}|r_{T,i} - m_{T,i}| \leq \eta_{T,i}B_{T-1} \leq \frac{B_{T-1}}{2B_{T-1}} = \frac{1}{2},$$

which meets the criterion for applying the mentioned inequality. By plugging inequality (22) into (21), we can further analyze the weights for all experts at time $T$:

$$\begin{aligned}
\sum_{i=1}^{N} w_{T+1,i} &\leq \sum_{i=1}^{N} \widetilde{w}_{T,i}\left(1 + \eta_{T,i}\left(\bar{r}_{T,i} - m_{T,i}\right)\right) + \sum_{i=1}^{N} \frac{1}{e}\left(\frac{\eta_{T,i}}{\eta_{T+1,i}} - 1\right) \\
&= \sum_{i=1}^{N} \widetilde{w}_{T,i}\left(1 - \eta_{T,i}m_{T,i}\right) + \sum_{i=1}^{N} \eta_{T,i}\widetilde{w}_{T,i}\bar{r}_{T,i} + \sum_{i=1}^{N} \frac{1}{e}\left(\frac{\eta_{T,i}}{\eta_{T+1,i}} - 1\right) \\
&\leq \sum_{i=1}^{N} \widetilde{w}_{T,i}\exp(-\eta_{T,i}m_{T,i}) + \sum_{i=1}^{N} \eta_{T,i}\widetilde{w}_{T,i}\bar{r}_{T,i} + \sum_{i=1}^{N} \frac{1}{e}\left(\frac{\eta_{T,i}}{\eta_{T+1,i}} - 1\right) \\
&= \sum_{i=1}^{N} w_{T,i} + \left(\sum_{j=1}^{N} \eta_{T,j}\widetilde{w}_{T,j}\right)\sum_{i=1}^{N} p_{T,i}\bar{r}_{T,i} + \sum_{i=1}^{N} \frac{1}{e}\left(\frac{\eta_{T,i}}{\eta_{T+1,i}} - 1\right) \\
&\leq \sum_{i=1}^{N} w_{T,i} + \sum_{i=1}^{N} \frac{1}{e}\left(\frac{\eta_{T,i}}{\eta_{T+1,i}} - 1\right),
\end{aligned}$$

where we apply $1 - x \leq \exp(-x)$ for any $x \in \mathbb{R}$, and the last inequality is by the assumption in the theorem statement that $\sum_{i=1}^{N} p_{t,i}\bar{r}_{t,i} \leq 0$ for any $t \in [T]$.

Now we are ready to upper bound $W_{T+1}$ in an inductive style:

$$\begin{aligned}
W_{T+1} = \sum_{i=1}^{N} w_{T+1,i} &\leq \sum_{i=1}^{N} w_{T,i} + \sum_{i=1}^{N} \frac{1}{e}\left(\frac{\eta_{T,i}}{\eta_{T+1,i}} - 1\right) = W_T + \sum_{i=1}^{N} \frac{1}{e}\left(\frac{\eta_{T,i}}{\eta_{T+1,i}} - 1\right) \\
&\leq W_1 + \sum_{t=1}^{T}\sum_{i=1}^{N} \frac{1}{e}\left(\frac{\eta_{t,i}}{\eta_{t+1,i}} - 1\right) = N + \sum_{t=1}^{T}\sum_{i=1}^{N} \frac{1}{e}\left(\frac{\eta_{t,i}}{\eta_{t+1,i}} - 1\right), \tag{23}
\end{aligned}$$

where the last inequality is by the induction. It remains to analyze the last term, the deviations of the learning rates. We present the following analysis tailored for the new learning rate setting, $\forall i \in [N]$:

$$\begin{aligned}
\frac{\eta_{T,i}}{\eta_{T+1,i}} - 1 &= \sqrt{\frac{1 + \sum_{t=1}^{T}(\bar{r}_{t,i} - m_{t,i})^2 + 4B_T^2}{1 + \sum_{t=1}^{T-1}(\bar{r}_{t,i} - m_{t,i})^2 + 4B_{T-1}^2}} - 1 \\
&= \sqrt{1 + \frac{4B_T^2 - 4B_{T-1}^2 + (\bar{r}_{T,i} - m_{T,i})^2}{1 + \sum_{t=1}^{T-1}(\bar{r}_{t,i} - m_{t,i})^2 + 4B_{T-1}^2}} - 1 \\
&\leq \frac{1}{2}\cdot\frac{4B_T^2 - 4B_{T-1}^2 + (\bar{r}_{T,i} - m_{T,i})^2}{1 + \sum_{t=1}^{T-1}(\bar{r}_{t,i} - m_{t,i})^2 + 4B_{T-1}^2} \qquad (\sqrt{1+x} \leq 1 + \tfrac{1}{2}x) \\
&= \frac{1}{2}\cdot\frac{\phi_{T,i}}{1 + 4B_0^2 + \sum_{t=1}^{T-1}\phi_{t,i}}. \qquad (\phi_{t,i} \triangleq 4B_t^2 - 4B_{t-1}^2 + (\bar{r}_{t,i} - m_{t,i})^2 \geq 0)
\end{aligned}$$

By Lemma 5 with the choices of $f(x) = 1/x, a_0 = 1 + 4B_0^2, a_t = \phi_{t,i}$ in the lemma statement, summing up the preceding inequality from $1$ to $T$ results in:

$$\sum_{t=1}^{T}\left(\frac{\eta_{t,i}}{\eta_{t+1,i}} - 1\right) \leq \frac{2B_T^2}{1 + 4B_0^2} + \frac{1}{2}\ln\left(1 + 4B_T^2 + \sum_{t=1}^{T}(\bar{r}_{t,i} - m_{t,i})^2\right) - \frac{1}{2}\ln(1 + 4B_0^2)$$

$$\leq \frac{2B_T^2}{1+4B_0^2} + \frac{1}{2}\ln\left(\frac{1+(T+4)B_T^2}{1+4B_0^2}\right).$$ (24)

Now combining (23) and (24), we can upper bound $\ln W_{T+1}$ as follows:

$$\ln W_{T+1} \leq \ln\left(1 + \frac{B_T^2}{1+4B_0^2} + \frac{1}{2e}\ln\left(\frac{1+TB_T^2}{1+4B_0^2}\right)\right) + \ln N.$$ (25)

In another direction, we lower bound $\ln W_{T+1} \geq \ln w_{T+1,i_\star}$ with an inductive argument:

$$\frac{1}{\eta_{T+1,i_\star}}\ln w_{T+1,i_\star} = \frac{1}{\eta_{T,i_\star}}\left(\ln w_{T,i_\star} + \eta_{T,i_\star}\bar{r}_{T,i_\star} - \eta_{T,i_\star}^2(\bar{r}_{T,i_\star} - m_{T,i_\star})^2\right)$$

$$= \frac{1}{\eta_{T,k}}\ln w_{T,i_\star} - \eta_{T,i_\star}(\bar{r}_{T,i_\star} - m_{T,i_\star})^2 + \bar{r}_{T,i_\star}$$

$$= \frac{1}{\eta_{1,i_\star}}\ln w_{1,i_\star} - \sum_{t=1}^T \eta_{t,i_\star}(\bar{r}_{t,i_\star} - m_{t,i_\star})^2 + \sum_{t=1}^T \bar{r}_{t,i_\star}$$

$$= -\sum_{t=1}^T \eta_{t,i_\star}(\bar{r}_{t,i_\star} - m_{t,i_\star})^2 + \sum_{t=1}^T \bar{r}_{t,i_\star}. \qquad (w_{1,i_\star}=1)$$

Rearranging the above equality with notice of Eq. (20), we have:

$$\sum_{t=1}^T r_{t,i_\star} \leq \sum_{t=1}^T \bar{r}_{t,i_\star} + B_T$$

$$\leq \sum_{t=1}^T \eta_{t,i_\star}(\bar{r}_{t,i_\star} - m_{t,i_\star})^2 + \frac{1}{\eta_{T+1,i_\star}}\left(\ln\left(1 + \frac{B_T^2}{1+4B_0^2} + \frac{1}{2e}\ln\left(\frac{1+TB_T^2}{1+4B_0^2}\right)\right) + \ln N\right) + B_T,$$

where the second term is in the order of

$$\mathcal{O}\left(\sqrt{B_T^2 + \sum_{t=1}^T (r_{t,i_\star} - m_{t,i_\star})^2} \cdot (\log(B_T + \log(TB_T)) + \log N)\right).$$

As for the first term, by applying Lemma 6, it can be bounded as follows:

$$\sum_{t=1}^T \eta_{t,i_\star}(\bar{r}_{t,i_\star} - m_{t,i_\star})^2 = \sum_{t=1}^T \frac{(\bar{r}_{t,i_\star} - m_{t,i_\star})^2}{\sqrt{1 + 4B_t^2 + \sum_{s=1}^{t-1}(\bar{r}_{s,i_\star} - m_{s,i_\star})^2}}$$

$$\leq \sum_{t=1}^T \frac{(\bar{r}_{t,i_\star} - m_{t,i_\star})^2}{\sqrt{1 + \sum_{s=1}^t (\bar{r}_{s,i_\star} - m_{s,i_\star})^2}} \leq 2\sqrt{1 + \sum_{t=1}^T (\bar{r}_{t,i_\star} - m_{t,i_\star})^2}$$

$$= 2\sqrt{1 + \sum_{t=1}^T \frac{B_{t-1}^2}{B_t^2}(r_{t,i_\star} - m_{t,i_\star})^2} \leq 2\sqrt{1 + \sum_{t=1}^T (r_{t,i_\star} - m_{t,i_\star})^2}.$$

Thus, the proof is complete. $\qquad\qquad\square$

### C.4 Proof of Theorem 4

*Proof.* We first decompose the static regret based on the performance of base-learner $i_\star$ into two parts as presented in Eq. (8).

The base-regret is guaranteed by the corresponding base-learner via Theorem 1 and Theorem 2. And we mainly focus on the analysis of the meta-regret by leveraging Theorem 6. First we are required to verify the condition that $\sum_{i\in[N]} p_{t,i}\bar{r}_{t,i} \leq 0$. Without loss of generality, we assume the 1-st base-learner is for convex functions. Recall that we set $r_{t,i} = \langle \nabla f_t(\mathbf{x}_t), \mathbf{x}_t - \mathbf{x}_{t,i} \rangle$ for strongly convex

functions learners $i \in [2, N]$, $r_{t,1} = f_t(\mathbf{x}_t) - f_t(\mathbf{x}_{t,1})$ for the convex function learner, and optimism $m_{t,i} = f_{t-1}(\mathbf{x}_t) - f_{t-1}(\mathbf{x}_{t,i})$ for each base-learner $i \in [N]$, therefore we have:

$$\sum_{i \in [N]} p_{t,i} \bar{r}_{t,i} = \left(1 - \frac{B_{t-1}}{B_t}\right)\left(f_{t-1}(\mathbf{x}_{t-1}) - \sum_{i \in [N]} p_{t,i} f_{t-1}(\mathbf{x}_{t,i})\right)$$

$$+ \left(p_{t,1}(f_t(\mathbf{x}_t) - f_t(\mathbf{x}_{t,1})) + \sum_{i \in [2,N]} p_{t,i}\langle \nabla f_t(\mathbf{x}_t), \mathbf{x}_t - \mathbf{x}_{t,i}\rangle\right)$$

$$\leq \left(1 - \frac{B_{t-1}}{B_t}\right)\left(\sum_{i \in [N]} p_{t,i} f_{t-1}(\mathbf{x}_{t,i}) - \sum_{i \in [N]} p_{t,i} f_{t-1}(\mathbf{x}_{t,i})\right)$$

$$+ \left(p_{t,1}\langle f_t(\mathbf{x}_t), \mathbf{x}_t - \mathbf{x}_{t,1}\rangle + \sum_{i \in [2,N]} p_{t,i}\langle \nabla f_t(\mathbf{x}_t), \mathbf{x}_t - \mathbf{x}_{t,i}\rangle\right) = 0.$$

Therefore, Theorem 6 is applicable in analyzing the meta-regret for universal online learning. In follows, we prove the regret bounds for convex functions and strongly convex functions respectively.

**Convex functions.** By Theorem 1, the base-regret is bounded by $\mathcal{O}(\sqrt{V_T})$, as for the meta-regret, by the setting of inputs and optimism, by Theorem 6, it is bounded by

$$\text{META-REG} \leq \mathcal{O}\left(\sqrt{\sum_{t=1}^{T}\left(\left(f_t(\mathbf{x}_t) - f_t(\mathbf{x}_{t,1})\right) - \left(f_{t-1}(\mathbf{x}_t) - f_{t-1}(\mathbf{x}_{t,1})\right)\right)^2 \cdot C_T + B_T \log B_T}\right)$$

$$= \mathcal{O}\left(\sqrt{\sum_{t=1}^{T}\left(\left(f_t(\mathbf{x}_t) - f_{t-1}(\mathbf{x}_t)\right) - \left(f_t(\mathbf{x}_{t,1}) - f_{t-1}(\mathbf{x}_{t,1})\right)\right)^2 \cdot C_T + B_T \log B_T}\right)$$

$$= \mathcal{O}\left(\sqrt{\sum_{t=1}^{T}\left(\langle \nabla f_t(\boldsymbol{\xi}_{t,1}) - \nabla f_{t-1}(\boldsymbol{\xi}_{t,1}), \mathbf{x}_t - \mathbf{x}_{t,1}\rangle\right)^2 \cdot C_T + B_T \log B_T}\right)$$

$$\leq \mathcal{O}\left(D\sqrt{\sum_{t=1}^{T}\sup_{\mathbf{x}\in\mathcal{X}}\|\nabla f_t(\mathbf{x}) - \nabla f_{t-1}(\mathbf{x})\|_2^2 \cdot C_T + B_T \log B_T}\right) = \mathcal{O}\left(\sqrt{V_T} \cdot C_T\right),$$

where we denote by $C_T = \mathcal{O}(\log(B_T + \log(B_T T)))$ and in the third line we apply the mean value theorem. Combining the base-regret and meta-regret together, we concludes that the static regret bound for convex functions is bounded by $\mathcal{O}(\sqrt{V_T} \cdot \log(B_T + \log(B_T T)))$, where $B_T = \mathcal{O}(\widehat{G}_{\max} D)$ with $\widehat{G}_{\max}$ denoting the maximum Lipschitz constant.

**Strongly convex functions.** For $\lambda_\star$-strong convex functions with $\lambda_\star \in [1/T, 1]$, by the construction of the curvature coefficient pool $\mathcal{H}$, there exists $i_\star \in [2, N]$ such that

$$\lambda_{i_\star} \leq \lambda_\star \leq 2\lambda_{i_\star}.$$

With this specific $i_\star$-th base-learner, the base-regret can be upper bounded by $\mathcal{O}((\log V_T)/\lambda_\star)$ by Theorem 2, up to a multiplicative constant of 2. The meta-regret can be bounded as follows:

$$\text{META-REG} \leq \sum_{t=1}^{T}\langle \nabla f_t(\mathbf{x}_t), \mathbf{x}_t - \mathbf{x}_{t,i_\star}\rangle - \sum_{t=1}^{T}\frac{\lambda_\star}{2}\|\mathbf{x}_t - \mathbf{x}_{t,i_\star}\|_2^2$$

$$\leq \mathcal{O}\left(\sqrt{\sum_{t=1}^{T}(\langle \nabla f_t(\mathbf{x}_t), \mathbf{x}_t - \mathbf{x}_{t,i_\star}\rangle - (f_{t-1}(\mathbf{x}_t) - f_{t-1}(\mathbf{x}_{t,i_\star})))^2 \cdot C_T} - \sum_{t=1}^{T}\frac{\lambda_\star}{2}\|\mathbf{x}_t - \mathbf{x}_{t,i_\star}\|_2^2 + B_T C_T\right)$$

$$\leq \mathcal{O}\left(\sqrt{\sum_{t=1}^{T}\langle \nabla f_t(\mathbf{x}_t), \mathbf{x}_t - \mathbf{x}_{t,i_\star}\rangle^2 + \langle \nabla f_{t-1}(\boldsymbol{\xi}_{t,i_\star}), \mathbf{x}_t - \mathbf{x}_{t,i_\star}\rangle^2 \cdot C_T} - \sum_{t=1}^{T}\frac{\lambda_\star}{2}\|\mathbf{x}_t - \mathbf{x}_{t,i_\star}\|_2^2 + B_T C_T\right)$$

$$\leq \mathcal{O}\left(\widehat{G}_{\max}\sqrt{\sum_{t=1}^{T}\|\mathbf{x}_t - \mathbf{x}_{t,i_\star}\|_2^2 \cdot C_T} - \sum_{t=1}^{T}\frac{\lambda_\star}{2}\|\mathbf{x}_t - \mathbf{x}_{t,i_\star}\|_2^2 + B_T C_T\right)$$

---

**Algorithm 3** Universal Gradient-Variation Online Learning under Global Smoothness

---

**Input:** curvature coefficient pools $\mathcal{H}_{sc}, \mathcal{H}_{exp}$, number of base-learners $N$, optimistic Adapt-ML-Prod [Wei et al., 2016] as the meta-algorithm, base-algorithms $\mathcal{A}_{cvx}, \mathcal{A}_{sc}, \mathcal{A}_{exp}$.

1: **Initialization**: Pass $N$ to the meta-algorithm, initialize a base-learner with $\mathcal{A}_{cvx}$; for $\lambda \in \mathcal{H}_{sc}$, initialize a base-learner with $\mathcal{A}_{sc}$; for $\alpha \in \mathcal{H}_{exp}$, initialize a base-learner with $\mathcal{A}_{exp}$.
2: **for** $t = 1$ **to** $T$ **do**
3:     Obtain $\boldsymbol{p}_t$ from meta-algorithm, $\mathbf{x}_{t,i}$ from each base-learner $i \in [N]$;
4:     Submit $\mathbf{x}_t = \sum_{i \in [N]} p_{t,i} \mathbf{x}_{t,i}$;
5:     Receive $\nabla f_t(\mathbf{x}_t)$ and send it to each base-learner for update;
6:     **For strongly convex and exp-concave functions learners**: set $r_{t,i} = \langle \nabla f_t(\mathbf{x}_t), \mathbf{x}_t - \mathbf{x}_{t,i} \rangle$;
7:     **For convex functions learner**: set $r_{t,1} = f_t(\mathbf{x}_t) - f_t(\mathbf{x}_{t,1})$;
8:     Send $\boldsymbol{r}_t$ to the meta-algorithm;
9:     Send $m_{t+1,1} = f_t(\mathbf{x}_t) - f_t(\mathbf{x}_{t,1})$ and $m_{t+1,i} = 0, i \in [2, N]$ to the meta-algorithm.
10: **end for**

---

$$\leq \mathcal{O}\left(\frac{\widehat{G}_{\max}^2 C_T^2}{\lambda_\star} + B_T \log B_T\right) = \mathcal{O}\left(\frac{\widehat{G}_{\max}^2 \log^2 B_T}{\lambda_\star} + B_T \log B_T\right),$$

where we use $\widehat{G}_{\max}$ to denote the maximum Lipschitz constant on the optimization trajectory, treat $\log \log T$ as constants, and $B_T = \mathcal{O}(\widehat{G}_{\max} D)$. In the fourth line, we again utilize the mean value theorem. The fifth line follows from Lemma 7, which ensures that:

$$|\langle \nabla f_{t-1}(\boldsymbol{\xi}_{t,i_\star}), \mathbf{x}_t - \mathbf{x}_{t,i_\star} \rangle| \leq \max \left\{ |\langle \nabla f_{t-1}(\mathbf{x}_{t,i_\star}), \mathbf{x}_t - \mathbf{x}_{t,i_\star} \rangle|, |\langle \nabla f_{t-1}(\mathbf{x}_t), \mathbf{x}_t - \mathbf{x}_{t,i_\star} \rangle| \right\},$$

thus, our result depends on the Lipschitz constant on the optimization trajectory. In the last step, we apply the AM-GM inequality. The above statements show that the meta-regret is bounded by constants. With the base-regret guarantee, the proof for strongly convex functions is complete. □

## C.5 A Simple Universal Algorithm under Global Smoothness

As a byproduct, our techniques can be used to design a simpler two-layer universal algorithm that achieves the optimal gradient-variation regret bounds for convex, strongly convex, and exp-concave functions simultaneously, thereby improving upon the results of Yan et al. [2023]. The crux involves leveraging the function-variation-to-gradient-variation technique for convex functions, and following the strategy of Zhang et al. [2022a] to demonstrate that the meta-regret is bounded by constants for strongly convex and exp-concave functions, at a cost of $\mathcal{O}(\log T)$ times function value queries per round. Given the global smoothness constant and the Lipschitz constant, our algorithm does not need to be Lipschitz-adaptive, thus suitable for more general optimism settings.

In Algorithm 3, we present this idea. In contrast to Algorithm 2, which is designed under the generalized smoothness, this algorithm in addition can guarantee gradient-variation bound for exp-concave functions. In below, we present the theoretical guarantees for Algorithm 3.

**Corollary 3.** *Under Assumption 2, and assuming the loss functions are $L$-smooth and $G$-Lipschitz, we set $N = 2\lceil \log_2 T \rceil + 1$. The curvature coefficient pools are defined as $\mathcal{H}_{sc} = \mathcal{H}_{exp} = \{2^{i-1}/T : i \in [(N-1)/2]\}$. By selecting suitable base-algorithms, for $\lambda, \alpha \in [1/T, 1]$, Algorithm 3 ensures the following results simultaneously:*

$$\text{REG}_T \leq \begin{cases} \mathcal{O}(\sqrt{V_T \cdot (\log \log T)}), & \text{(convex)}, \\ \mathcal{O}\left(\frac{1}{\lambda} \log V_T + \frac{G^2 \log \log T}{\lambda}\right), & (\lambda\text{-strongly convex}), \\ \mathcal{O}\left(\frac{d}{\alpha} \log V_T + \frac{\log \log T}{\alpha}\right), & (\alpha\text{-exp-concave}). \end{cases}$$

*Proof.* When assuming the Lipschitz constant $G$, optimistic Adapt-ML-Prod [Wei et al., 2016] can ensure the meta-regret bounded by $\mathcal{O}(\sqrt{\sum_{t=1}^{t}(r_{t,i} - m_{t,i})^2 \cdot \log \log T})$, thus the dependence of logarithmic terms is improved compared to Theorem 4. The proofs for convex functions and strongly convex functions are nearly identical to the proofs for Theorem 4 in Appendix C.4; thus, we omit them here. We highlight the importance of the function-variation-to-gradient-variation technique, bounding the meta-regret of order $\mathcal{O}(\sqrt{V_T \cdot \log \log T})$ without the cancellation-based analysis.

Next, we show that the meta-regret for $\alpha_\star$-exp-concave functions is bounded by a constant. By the construction of the curvature pool $\mathcal{H}_{\exp}$, there exists base-learner $i_\star$ with the input curvature $\alpha_{i_\star}$ satisfying that $\alpha_{i_\star} \leq \alpha_\star \leq 2\alpha_{i_\star}$. Decompose the regret against this specific base-learner and by the definition of exp-concave functions, we have:

$$\text{META-REG} = \sum_{t=1}^{T} f_t(\mathbf{x}_t) - f_t(\mathbf{x}_{t,i_\star}) \leq \sum_{t=1}^{T} \langle \nabla f_t(\mathbf{x}_t), \mathbf{x}_t - \mathbf{x}_{t,i_\star} \rangle - \frac{\alpha}{2} \langle \nabla f_t(\mathbf{x}_t), \mathbf{x}_t - \mathbf{x}_{t,i_\star} \rangle^2$$

$$\leq \mathcal{O}\left( \sqrt{\sum_{t=1}^{T} (r_{t,i_\star} - m_{t,i_\star})^2 \cdot \log\log T - \frac{\alpha}{2} \langle \nabla f_t(\mathbf{x}_t), \mathbf{x}_t - \mathbf{x}_{t,i_\star} \rangle^2} \right)$$

$$= \mathcal{O}\left( \sqrt{\sum_{t=1}^{T} (\langle \nabla f_t(\mathbf{x}_t), \mathbf{x}_t - \mathbf{x}_{t,i_\star} \rangle)^2 \cdot \log\log T - \frac{\alpha}{2} \langle \nabla f_t(\mathbf{x}_t), \mathbf{x}_t - \mathbf{x}_{t,i_\star} \rangle^2} \right) \leq \mathcal{O}\left( \frac{\log\log T}{\alpha} \right),$$

where the second-to-last line follows from the settings that $r_{t,i_\star} = \langle \nabla f_t(\mathbf{x}_t), \mathbf{x}_t - \mathbf{x}_{t,i_\star} \rangle$ and $m_{t,i_\star} = 0$, and we apply the AM-GM inequality in the final step. Therefore, by choosing the base-algorithm that ensures the regret bound of $\mathcal{O}(\frac{d}{\alpha_\star} \log V_T)$, we complete the proof for this theorem. $\qquad\square$

## D   Omitted Details for Section 4

This section provides the omitted proofs for our two applications.

### D.1   Proof of Corollary 1

*Proof.* We prove this theorem in a black-box manner, thanks to the gradient-variation bound we derive. By Theorem 4, for convex functions, Algorithm 2 ensures:

$$\text{REG}_T \leq \mathcal{O}\left( \sqrt{\sum_{t=1}^{T} \sup_{\mathbf{x} \in \mathcal{X}} \|\nabla f_t(\mathbf{x}) - \nabla F_t(\mathbf{x})\|_2^2 + \sum_{t=2}^{T} \sup_{\mathbf{x} \in \mathcal{X}} \|\nabla F_t(\mathbf{x}) - \nabla F_{t-1}(\mathbf{x})\|_2^2} \right),$$

where we decompose $\|\nabla f_t(\mathbf{x}) - \nabla f_{t-1}(\mathbf{x})\|_2^2$ as follows:

$$\mathcal{O}\left( \|\nabla f_t(\mathbf{x}) - \nabla F_t(\mathbf{x})\|_2^2 + \|\nabla F_t(\mathbf{x}) - \nabla F_{t-1}(\mathbf{x})\|_2^2 + \|\nabla F_{t-1}(\mathbf{x}) - \nabla f_{t-1}(\mathbf{x})\|_2^2 \right). \quad (26)$$

Finally, by taking expectation on both sides and leveraging the concavity of the square root, we have:

$$\mathbb{E}[\text{REG}_T] \leq \mathcal{O}\left( \sqrt{\sum_{t=1}^{T} \mathbb{E}\left[ \sup_{\mathbf{x} \in \mathcal{X}} \|\nabla f_t(\mathbf{x}) - \nabla F_t(\mathbf{x})\|_2^2 \right] + \sum_{t=2}^{T} \mathbb{E}\left[ \sup_{\mathbf{x} \in \mathcal{X}} \|\nabla F_t(\mathbf{x}) - \nabla F_{t-1}(\mathbf{x})\|_2^2 \right]} \right),$$

which is in order of $\mathcal{O}\left( \sqrt{\Sigma_I^2} + \sqrt{\tilde{\sigma}_I^2} \right)$.

For strongly convex functions, as shown in Eq. (13) in the proof of Theorem 2, the multiplicative factor $\widehat{G}_{\max}$ can be replaced by a more refined factor $\max_{t \in [T]} \|\nabla f_t(\mathbf{x}_t) - \nabla f_{t-1}(\mathbf{x}_t)\|_2$. Then Algorithm 2 can ensure the following bound for strongly convex functions:

$$\text{REG}_T \leq \mathcal{O}\left( \max_{t \in [T]} \|\nabla f_t(\mathbf{x}_t) - \nabla f_{t-1}(\mathbf{x}_t)\|_2^2 \cdot \log\left( \sum_{t=2}^{T} \sup_{\mathbf{x} \in \mathcal{X}} \|\nabla f_t(\mathbf{x}) - \nabla f_{t-1}(\mathbf{x})\|_2^2 \right) \right).$$

By applying a similar argument as in Eq. (26) to decompose the gradient variation, taking the expectation on both sides, and leveraging the concavity of the logarithm, we conclude the proof. $\qquad\square$

### D.2   Proof of Corollary 2

*Proof.* The step sizes for optimistic OMD in (10) is $\eta_t = \min\{D, \ \min_{s \in [t]} \frac{1}{4\ell_{s-1}(2\|F_{s-1}(\widehat{\mathbf{z}}_s)\|_2)}\}$. By the convexity and the concavity for the objective function $f(\cdot, \cdot)$, for any $\mathbf{z} = (\mathbf{x}, \mathbf{y}) \in \mathcal{Z}$, we can linearize the gap for an $\varepsilon$-approximate solution as $f(\bar{\mathbf{x}}_T, \mathbf{y}) - f(\mathbf{x}, \bar{\mathbf{y}}_T) \leq \frac{1}{T} \sum_{t=1}^{T} \langle F(\mathbf{z}_t), \mathbf{z}_t - \mathbf{z} \rangle$.

Following the proof of Lemma 2, we can demonstrate that optimistic OMD in (10) ensures:

$$\sum_{t=1}^{T} \langle F(\mathbf{z}_t), \mathbf{z}_t - \mathbf{z} \rangle \leq \mathcal{O}\left( \frac{\max_{t \in [T]} \|\mathbf{z}_t - \mathbf{z}\|_2^2}{\eta_T} + \sum_{t=1}^{T} \eta_t \|F(\mathbf{z}_t) - F(\widehat{\mathbf{z}}_t)\|_2^2 - \sum_{t=1}^{T} \frac{1}{\eta_t} \|\mathbf{z}_t - \widehat{\mathbf{z}}_t\|_2^2 \right).$$

The first term on the right-hand side is in the order of $\mathcal{O}(\widehat{L}_{\max} D^2)$ by the step size configuration, where $\widehat{L}_{\max}$ denotes the maximum smoothness constant on the optimization trajectory. By the $\ell$-smoothness in Definition 2, the second term can be bounded as $\eta_t \|F(\mathbf{z}_t) - F(\widehat{\mathbf{z}}_t)\|_2^2 \leq \mathcal{O}(\eta_t \ell_{s-1}(2\|F_{s-1}(\widehat{\mathbf{z}}_s)\|_2)^2 \cdot \|\mathbf{z}_t - \widehat{\mathbf{z}}_t\|_2^2)$, which can be further cancelled by the negative terms. Therefore, $\sum_{t=1}^{T} \langle F(\mathbf{z}_t), \mathbf{z}_t - \mathbf{z} \rangle$ is bounded by a constant, thus, the convergence rate to an $\varepsilon$-approximate solution is $\mathcal{O}(1/T)$, implying a fast convergence rate. $\qquad\square$

## E  Supporting Lemmas

### E.1  Lemmas for Optimistic OMD

We first present the generic lemma for optimistic OMD with dynamic regret [Zhao et al., 2024], which encompasses the standard regret by setting $\mathbf{u}_1 = ... = \mathbf{u}_T = \mathbf{x}_\star \in \arg\min_{\mathbf{x} \in \mathcal{X}} \sum_{t=1}^{T} f_t(\mathbf{x})$.

**Lemma 3** (Theorem 1 of Zhao et al. [2024]). *Optimistic OMD specialized at Eq. (3) satisfies*

$$\sum_{t=1}^{T} \langle \nabla f_t(\mathbf{x}_t), \mathbf{x}_t - \mathbf{u}_t \rangle \leq \sum_{t=1}^{T} \langle \nabla f_t(\mathbf{x}_t) - M_t, \mathbf{x}_t - \widehat{\mathbf{x}}_{t+1} \rangle + \sum_{t=1}^{T} \left( \mathcal{D}_{\psi_t}(\mathbf{u}_t, \widehat{\mathbf{x}}_t) - \mathcal{D}_{\psi_t}(\mathbf{u}_t, \widehat{\mathbf{x}}_{t+1}) \right)$$

$$- \sum_{t=1}^{T} \left( \mathcal{D}_{\psi_t}(\widehat{\mathbf{x}}_{t+1}, \mathbf{x}_t) + \mathcal{D}_{\psi_t}(\mathbf{x}_t, \widehat{\mathbf{x}}_t) \right),$$

*where $\mathbf{u}_1, \ldots, \mathbf{u}_T \in \mathcal{X}$ are arbitrary comparators in the feasible domain.*

The next lemma, known as the stability lemma, establishes an upper bound on the proximity between successive decisions in terms of the gradient utilized for updates.

**Lemma 4** (Proposition 7 of Chiang et al. [2012]). *Consider the following two updates: (i) $\mathbf{x} = \arg\min_{\mathbf{x} \in \mathcal{X}} \{\langle \mathbf{g}, \mathbf{x} \rangle + \mathcal{D}_\psi(\mathbf{x}, \mathbf{c})\}$, and (ii) $\mathbf{x}' = \arg\min_{\mathbf{x} \in \mathcal{X}} \{\langle \mathbf{g}', \mathbf{x} \rangle + \mathcal{D}_\psi(\mathbf{x}, \mathbf{c})\}$. When the regularizer $\psi : \mathcal{X} \mapsto \mathbb{R}$ is $\lambda$-strongly convex function with respect to norm $\|\cdot\|$, we have $\lambda \|\mathbf{x} - \mathbf{x}'\| \leq \|\mathbf{g} - \mathbf{g}'\|_*$.*

### E.2  Self-Confident Tuning Lemmas

In this part, we provide some useful lemmas when analyzing the self-confident tuning strategy.

**Lemma 5** (Extension of Lemma 14 Gaillard et al. [2014]). *Let $a_0 > 0$ and $a_t \in [0, B]$ be real numbers for all $t \in [T]$ and let $f : (0, +\infty) \mapsto [0, +\infty)$ be a nonincreasing function. Then $\sum_{t=1}^{T} a_t f\left( \sum_{s=0}^{t-1} a_s \right) \leq B \cdot f(a_0) + \int_{a_0}^{\sum_{t=0}^{T} a_t} f(u) \mathrm{d}u$.*

**Lemma 6** (Lemma 3.5 of Auer et al. [2002]). *Let $a_1, \ldots, a_T$ and $\delta$ be non-negative real numbers. Then $\sum_{t=1}^{T} \frac{a_t}{\sqrt{\delta + \sum_{s=1}^{t} a_s}} \leq 2\left( \sqrt{\delta + \sum_{t=1}^{T} a_t} - \sqrt{\delta} \right)$.*

### E.3  Technical Lemma

**Lemma 7.** *Let $f : \mathcal{X} \mapsto \mathbb{R}$ be a convex, twice differentiable function. Then for any $\mathbf{x}, \mathbf{y} \in \operatorname{int} \mathcal{X}$ and $\lambda \in [0, 1]$, we have that:*

$$|\langle \nabla f(\lambda \mathbf{x} + (1 - \lambda)\mathbf{y}), \mathbf{x} - \mathbf{y} \rangle| \leq \max \{|\langle \nabla f(\mathbf{x}), \mathbf{x} - \mathbf{y} \rangle|, |\langle \nabla f(\mathbf{y}), \mathbf{x} - \mathbf{y} \rangle|\}.$$

*Proof.* Define a function that $\phi(t) = f(t\mathbf{x} + (1 - t)\mathbf{y})$. For this univariate convex function, we have $\phi'(t) = \langle \nabla f(t\mathbf{x} + (1 - t)\mathbf{y}), \mathbf{x} - \mathbf{y} \rangle$. By the convexity of $\phi(t)$, there is $|\phi'(t)| \leq \max \{|\phi'(1)|, |\phi'(0)|\}$, concluding the proof. $\qquad\square$

