# OpenReview forum: "Gradient-Variation Online Learning under Generalized Smoothness"
_NeurIPS.cc/2024/Conference — NeurIPS 2024 poster_

### Official Review · Reviewer_YPdL · 2024-06-28

**Soundness:** 4
**Presentation:** 3
**Contribution:** 3
**Rating:** 7
**Confidence:** 3

**Summary:**

This paper contributes gradient-variation extensions of several online learning guarantees to
a generalized smoothness setting. Under a more general smoothness assumption,
the paper first provides an algorithm which achieves a $O(\sqrt{V_T})$
gurantee, where $V_T=\sum_{t} \sup_{x}\\|\nabla f_t(x)-\nabla f_{t-1}(x)\\|^2$, as well as a $O(\log{V_T})$ guarantee for
strongly-convex functions.
The result is then extended to a universal guarantee, which holds
without knowledge of whether the losses are strongly convex or just convex.
The approach is also extended to provide an analogous strongly-adaptive guarantee and a $O(\sqrt{V_TP_T})$ dynamic regret guarantee.

**Strengths:**

The generalized smoothness assumption is nice. The paper reads well and the technical development easy to follow. Results seem novel.

**Weaknesses:**

The assumption that the learner has prior knowledge of and can query the smoothness function arbitrarily seems very strong

The method requires multiple-query gradient access

The approach requires prior knowledge of $\min_x f_t(x)$, which makes the problem quite a bit easier. One of the difficult things
that arises in these non-globally-bounded settings is that the learner doesn't have a proper frame-of-reference for how large
losses and gradients really are; giving the learner access to $\min_x f_t(x)$ provides a very strong version of such a frame-of-reference

**Questions:**

- "However, to ensure the theoretical results are valid, we assume that there exist finite but unknown upper bounds G and L for the Lipschitzness and smoothness following the discussion in Jacobsen and Cutkosky [35]."
  - What discussion is being referred to here specifically?

**Limitations:**

- The limitation in terms of a potential exp-concave result is stated explicitly in Section 2.3
- The assumption that $\min_x f_t(x)$ is required is pointed out explicitly. Although it it's significance
  is a bit downplayed.
- Multiple-query gradient access is also pointed out in the conclusion

---

> ### Author Rebuttal · Authors · 2024-08-05
>
> Thank you for your valuable feedback! In the following, we will answer your question and respond to your concerns regarding the assumptions.
>
> ---
>
> **Q1**. "However, to ensure the theoretical results are valid, we assume that there exist finite but unknown upper bounds G and L for the Lipschitzness and smoothness following the discussion in Jacobsen and Cutkosky [35]." What discussion is being referred to here specifically?"
>
> **A1**. Thanks for the question. We refer to the discussion on the finiteness of the Lipschitz constant and the smoothness constant in the right column of page 2 in Jacobsen and Cutkosky [35], as restated below: "Importantly, note that in all of these prior works there is an implicit assumption that there exists a uniform bound such that $ G \geq \\|\nabla \ell_t(w)\\|$ for any $w\in \mathcal{W}$ — even if it is not known in advance. Otherwise, the terms $G_T = \max_t \\|g_t\\|$ can easily make any of the aforementioned regret guarantees vacuous."
>
> ---
>
> **Q2**. "The assumption that the learner has prior knowledge of and can query the smoothness function arbitrarily seems very strong." "The method requires multiple-query gradient access."
>
> **A2**. Thanks for the comments. We will respond to these two comments together since, for generalized smoothness, querying gradients is directly related to estimating smoothness.
>
> - **Prior knowledge of the smoothness function**: For the general form of generalized smoothness, this query requirement is arguably strong. Technically, given the weaker notion of smoothness we are working with, obtaining certain *local* information is essential to properly adapt to the optimization trajectory. On the other hand, in the case of the commonly studied $(L_0, L_1)$-smoothness, which is one representative instance of generalized smoothness often observed in neural network optimization, this query becomes inexpensive. This is because we can directly calculate the local smoothness $L_t = L_0+L_1*\\| \nabla f_t(x_t)\\|$ by definition provided that $L_0$ and $L_1$ are known (they are usually empirically estimated on the fly for NN optimization).
> - **Multiple-query gradient access**: When assuming the global smoothness, the information (gradients, smoothness) of the combined decision $x_t = \sum_{i=1}^N p_{t,i}x_{t,i}$ can be shared to tune base-learners. Consequently, previous works have developed algorithms by thoroughly utilizing this shared information.  However, when considering the generalized smoothness, *the estimation of the smoothness is highly correlated to the optimization trajectory and is valid only within a local region*. Utilizing the gradient and smoothness at point $x_t$ leads to improper tuning of the base learners, which in turn affects the generation of the subsequent decision $x_{t+1}$, complicating the tuning process. Therefore, we have to decouple the design of meta and base levels, and require multiple-query gradient access to estimate the smoothness for each base-learner. Reducing the number of gradient queries is definitely an important future work. Thanks for pointing it out!
> ---
> **Q3**. "The approach requires prior knowledge of $\min_x f_t(x)$, which makes the problem quite a bit easier."
>
> **A3**. Thanks for the comment. We need to clarify that we only require knowledge of $\min_x f_t(x)$ when developing the universal method; it is not required for non-stationary online algorithms. Removing this assumption is important for future work, but currently, we have to work with it. Below, we provide explanations for the technical necessity and justifications.
>
> - **Technical reason for requiring this assumption**: The assumption for the $\min_x f_t(x)$ arises from the heterogeneous inputs and the binary search operation in the meta-algorithm. To accommodate the heterogeneous inputs, we need to search $p_t$ such that $f_t(\sum_{i=1}^N p_{t,i}x_{t,i})$ satisfies some properties. Here, we require the lower bound of $f_t(x)$ to facilitate the binary search operation. In contrast, when developing the adaptive regret minimization method, it is acceptable to search $p_t$ such that $\sum_{i=1}^N p_{t,i} f_t(x_{t,i})$ satisfies the same properties. Notice that, in this case, the range of $\sum_{i=1}^N p_{t,i} f_t(x_{t,i})$ is $[\min_{i} f_t(x_{t,i}), \max_{i} f_t(x_{t,i})]$ as $p_t$ is from simplex.
> - **Justification and support for this assumption**: As mentioned in the paper, in the typical Empirical Risk Minimization (ERM) setting, this lower bound certification is mild, as recent works in optimization suggest that $0$ can naturally serve as a lower bound for $f_t(x)$ [3, discussion below Theorem 1]. Other recent works in parameter-free stochastic optimization also necessitate a lower bound on function values, concretely, the learning rate tuning in Theorem 1 of [4] and the inputs in Algorithm 1 of [5].
>
> We will consider how to remove this assumption, which is definitely an important yet challenging future work to address.
>
> ---
>
> **References:**
>
> [1] Why gradient clipping accelerates training: A theoretical justification for adaptivity, ICLR'20.
>
> [2] Robustness to unbounded smoothness of generalized signSGD, NeurIPS'22.
>
> [3] Revisiting the Polyak step size, arXiv'19.
>
> [4] How free is parameter-free stochastic optimization?, ICML'24.
>
> [5] Tuning-free stochastic optimization, ICML'24.
>
> [Paper ref 35] Unconstrained online learning with unbounded losses, ICML'23.

---

> > ### Comment · Reviewer_YPdL · 2024-08-07
> >
> > Thank you for the detailed response! My main concerns were addressed well so I have raised my score.
> >
> > **A1:** Makes sense; I think it would be helpful to the reader if you added more specifics as to which discussion you're referring to, since they might not be familiar with that paper

---

> > > ### Author Response · Authors · 2024-08-10
> > >
> > > Thank you for your comment! We will improve the presentation to make this point clearer.

---

### Official Review · Reviewer_NEvk · 2024-07-07

**Soundness:** 3
**Presentation:** 2
**Contribution:** 3
**Rating:** 6
**Confidence:** 3

**Summary:**

Problem: The paper studies the OCO problem under the generalized smoothness assumption, i.e. at each time $t$, $f_{t}$ satisfies $\|| \nabla ^ {2} f_{t} (x) \|| \le \ell_{t}(\|| \nabla f_{t}(x) \||)$ for all $x \in \mathcal{X}$, where $\ell_{t}$ is a positive non-decreasing function. In addition to the information model in OCO, the algorithm can query $\ell_{t}$ at any local point $x \in \mathcal{X}$. This condition subsumes global smoothness, i.e. $\ell \le L$ is upper bounded globally, and $(L_{0},L_{1})$ smoothness, i.e. $\ell(x) = L_0 + L_1 x$; which have been studied in the prior works. The authors obtain gradient-based variation bounds for several settings of interest (see contributions below).

Contributions:
1) First, the authors extend the classic optimistic OMD algorithm to handle generalized smoothness and obtain an optimal $\mathcal{O}(\log V_{T})$ rate for strongly convex and $\mathcal{O}(\sqrt{V_{T}})$ static regret for convex functions (ref. Theorem 1, 2). However, these rates are first derived assuming that the algorithm knows the curvature of the $f_{t}$'s, i.e. whether all the $f_{t}$'s are convex or strongly convex.

2) To circumvent the curvature information, the authors devise an algorithm that is adaptive to the curvature of the functions $f_{t}$'s (referred to as universal online learning -- the goal is to obtain an algorithm with simultaneous guarantees for convex and strongly convex functions). The proposed algorithm obtains the optimal $\mathcal{O}(\sqrt{V_T}), \mathcal{O}(\log V_T)$ static regret bounds for convex, strongly convex functions respectively (ref. Theorem 4).

3) Finally, the authors consider stronger regret measures, i.e. interval regret and dynamic regret (referred to as non-stationary online learning), and propose an algorithm with the optimal gradient variation-based regret guarantees (ref. Theorem 5, 6).

**Strengths:**

The paper is solid and shows that gradient variation-based bounds can be derived under a more general smoothness assumption. Prior works focused on global smoothness, or $(L_{0}, L_{1})$ smoothness, and did not consider some or the other regret metrics as considered in the paper, e.g. the authors mention that an adaptive regret guarantee was not obtained even in the context of global smoothness.

While this might seem doable, the authors very well mention the key challenges in obtaining the desired algorithms, e.g. for contribution (2) above, section 3.2 does a great job of highlighting the key challenges. I liked this section and the authors spent considerable effort explaining why existing meta algorithms, e.g. McMwC-Master algorithm (Chen et al.) do not directly work since they cannot handle heterogeneous inputs.

The function-to-gradient variation technique is simple and seems pretty useful since it allows us to decouple the meta-algorithm and the base learners and explicitly aim toward obtaining a meta-algorithm with the desired guarantee. It helps avoid the seemingly messier cancellation-based tricks as done by Yan et al.

**Weaknesses:**

1) Section 4 is presented in a rush. Like section 3.1, it would be quite beneficial to have some more discussion on the algorithm and potential challenges in deriving non-stationary regret guarantees. The authors claim that their proposed algorithm is the first to obtain gradient variation-based non-stationary regret guarantees under the generalized smoothness assumption, whereas prior works did not obtain similar guarantees under the global smoothness assumption. The authors should have at least backed this with the potential difficulties faced by existing approaches. I recommend that the authors cut down some earlier discussions, such as the discussion about exp-concave functions, which in my opinion is not very relevant, given that the entire paper is about convex and strongly convex functions.

2) While the paper shows interesting results, I feel the key ideas are borrowed from existing works. While this is fine, in some places the authors failed to mention the potential technical challenges in the analysis. This is the most applicable to section 3. The authors mention that the proposed algorithm can be analyzed similarly to the prod-style algorithms. Are there technical difficulties in directly incorporating the analysis of prod-style algorithms into Algorithm 1?

Some minor typos that I found:

1) Line 38: "stochatsic -> stochastic"
2) Lemma 1: Remove the $\ell_t$

**Questions:**

1) The discussion towards the end of page 3 is confusing to me. The authors mention that assuming the Lipschitzness of $f_{t}$ is not reasonable since the bound on $\|| \nabla ^ {2} f(x) \||$ can be obtained from knowing $\ell$ directly, and the function is globally smooth. However, the bounds presented in the Theorems are a function of the Lipschitzness parameters. I would appreciate clarification from the authors.

2) In the Equation after Line 384, $\xi_{t, i}$ is not necessarily same for $f_t$ and $f_{t - 1}$ since the Mean Value Theorem only guarantees the existence of a $\xi$. With $\xi_t$ and $\xi_{t-1}$ being possibly different, I don't think the next inequality follows immediately. Is that a typo or am I missing something?

**Limitations:**

The authors adequately addressed the limitations.

---

> ### Author Rebuttal · Authors · 2024-08-05
>
> Thank you for your careful review and very constructive comments. We will revise the paper according to your suggestions. In the following, we address each of your technical questions.
>
> ---
>
> **Q1**. "The discussion towards the end of page 3 is confusing to me....the bounds presented in the Theorems are a function of the Lipschitzness parameters."
>
> **A1.** Thanks for the question. We make the following clarifications.
>
> - First, we do *not* assume the prior knowledge of the Lipschitz constant as an algorithmic input. Otherwise, with such foreknowledge, we could directly compute a global smoothness constant, reducing the problem to OCO with global smoothness.
> - Second, we do assume *finite* Lipschitz constants during the learning process, even though this finite upper bound is *unknown* to the algorithm. Without this assumption, the regret bounds would become vacuous. The Lipschitz constants presented in our theorems are evaluated on the fly, which demonstrates the Lipschitz-adaptivity of our algorithms when exploiting generalized smoothness.
> - We emphasize that the above assumption is fundamental and shared in OCO relating to the unbounded quantities; also refer to prior discussions [1] (Page 2, right column) "Importantly, note that in all of these prior works there is an implicit assumption that there exists a uniform bound such that $ G ≥ \\|\nabla \ell_t(w)\\|$ for any $w\in \mathcal{W}$ — even if it is not known in advance. Otherwise, the terms $G_T = \max_t \\|g_t\\|$ can easily make any of the aforementioned regret guarantees vacuous".
>
> We will improve the presentation to avoid any confusion on this point in the revised version.
>
> ---
>
> **Q2**. "With $\xi_t$ and $\xi_{t-1}$ being possibly different, I don't think the next inequality follows immediately. Is that a typo or am I missing something?"
>
> **A2**. Sorry for this confusion. The inequality is indeed correct, and we provide derivations for clarity. One can introduce a function $F_t(x) = f_t(x) - f_{t-1}(x)$, then the "function-variation quantity" is essentially $F_t(x_{t,i}) - F_t(x_{\text{ref}})$. Applying the Mean Value Theorem to $F_t$ yields
>
> $$F_t(x_{t,i}) - F_t(x_{\text{ref}}) = \langle \nabla F_t(\xi_{t,i}), x_{t,i} - x_{\text{ref}} \rangle = \langle \nabla f_t(\xi_{t,i}) - \nabla f_{t-1}(\xi_{t,i}), x_{t,i} - x_{\text{ref}} \rangle,$$
>
> where $\xi_{t,i}$ is a point guaranteed to lie between $x_t$ and $x_{\text{ref}}$.
>
> We will make it clear in the next version. Thanks!
>
> ----
>
> **Q3**.  "Section 4 is presented in a rush......prior works did not obtain similar guarantees (gradient variation-based non-stationary regret) under the global smoothness assumption.....The authors should have at least backed this with the potential difficulties faced by existing approaches....I recommend that the authors cut down some earlier discussions...."
>
> **A3**. Thanks for the suggestions! We will reserve space to incorporate additional details for Section 4 (especially given that one additional page is allowed in the final version).
>
> We briefly highlight difficulties faced by existing approaches in achieving gradient-variation adaptive regret. To the best of our understanding, previous efforts mainly focus on "cancellation-based analysis" to attain gradient variations. However, minimizing adaptive regret requires the meta-algorithm to accommodate the sleeping-expert mechanism, which significantly complicated the cancellation-based arguments due to the impact of varying numbers of active base-learners on negative terms. In contrast, our approach leverages the function-variation-to-gradient-variation conversion and thus can avoid this issue.
>
> ----
>
> **Q4**. "Are there technical difficulties in directly incorporating the analysis of prod-style algorithms into Algorithm 1?"
>
> **A4**. There are two key algorithmic modifications in Algorithm 1 compared to standard Prod-style algorithms, which have introduced technical difficulties.
>
> - **Clipping operation**: Due to the Lipschitz-adaptive requirement of meta-algorithm, we introduce a clipping operation [2] into the prod-style algorithm with optimism. This incorporation requires to *carefully design the optimism*. We demonstrate that the condition $\langle p_t, m_t \rangle \leq 0$ is essential for achieving regret bounds scaling with  $\sum_{t=1}^T (r_{t,i} - m_{t,i})^2$, which further leads to a new optimism design in the universal method.
> - **Learning rate setting**: We introduce a novel non-increasing learning rate setting, as opposed to prior Lipschitz-adaptive algorithms that use a fixed learning rate with restarts. This simplified and more practical learning rate setting requires sophisticated analysis. Specifically, the addition of $B_t^2$ in the denominator of learning rates is *novel* to prod-style algorithms. It not only *ensures* the key condition of $\eta_{t,i} |\bar{r}_{t,i} - m_{t,i}| \leq 1/2$, but also *removes* the threshold on learning rates commonly used in previous prod-style algorithms [3, 4], which guarantees that $\eta_{t,i}/\eta_{t+1,i}$ can still be well controlled (as analyzed at Line 899 in Appendix D.5).
>
> To summarize, extending prod-style algorithms with optimism to be Lipschitz-adaptive requires novel ingredients. We will add discussions to highlight those potential technical challenges.
>
> ---
>
> We will revise the paper to enhance our presentation's clarity and avoid confusion. Please consider updating the score if our clarifications have properly addressed your concern. Thank you!
>
> ---
>
> **References:**
>
> [1] Unconstrained online learning with unbounded losses, ICML'23.
>
> [2] Artificial constraints and hints for unbounded online learning, COLT'19.
>
> [3] A second-order bound with excess losses, COLT'14.
>
> [4] Tracking the best expert in non-stationary stochastic environments, NIPS'16.

---

> > ### Comment · Reviewer_NEvk · 2024-08-10
> > **Response to the Author Rebuttal**
> >
> > I thank the authors for their reply and appreciate their effort in the rebuttal.
> >
> > Response to A1: This makes sense to me now. I agree with the assumption that there exists a constant such that $||\nabla \ell_{t}(w)|| \le G$ for all $w \in \mathcal{W}$, is quite common in the online learning literature.
> >
> > Response to A2: Thanks for the clarification.
> >
> > Response to A3 + A4: I am convinced that the authors do propose some interesting techniques to get around the difficulties faced by existing works.
> >
> > Based on the author's responses, I have now increased my score to 6. However, please make sure to reorganize section 4 in the subsequent revisions of the paper.

---

> > > ### Author Response · Authors · 2024-08-10
> > >
> > > Thanks for your helpful comments! We will carefully revise the paper to ensure that key ideas and main results are clearly and properly presented in the final version.

---

### Official Review · Reviewer_ian3 · 2024-07-12

**Soundness:** 3
**Presentation:** 2
**Contribution:** 3
**Rating:** 6
**Confidence:** 2

**Summary:**

The paper aims to provide gradient-variation regret bound when only a generalized smoothness condition holds. It considers the optimistic mirror descent algorithm. It further designed a meta-algorithm, which can be Lipscthiz adaptive. Lastly, it considers adaptive regret and dynamic regret.

**Strengths:**

(1) Compared with existing literature, the work relaxed the smoothness assumption.

(2) The meta-algorithm to achieve universality seems novel to me.

**Weaknesses:**

The clarity and effectiveness of the paper could be improved by condensing the extensive paragraphs that discuss "challenges." Instead, it would be beneficial to expand the remarks on how the findings compare with existing results following the theorems. This would provide readers with a clearer understanding of the paper's contributions to the current body of knowledge.

**Questions:**

1. The paper addresses the generalized smoothness condition and frequently mentions the potential for unbounded gradients and smoothness. However, Assumption 2 posits a bounded domain. Does this imply that the gradients and smoothness are also bounded? In reference [7], such boundedness does not appear to be a requirement.

2. On page 5, line 203, the paper asserts that the complexity \(O(D\sqrt{V_T} + LD^2)\) aligns with the optimal constant dependency. It would be helpful if the authors could specify where exactly in reference [7] this lower bound is discussed.

---

> ### Author Rebuttal · Authors · 2024-08-05
>
> Thank you for your valuable comments.  We will revise our paper to highlight the contributions according to your suggestions. Below, we will address your questions.
>
> ---
>
> **Q1**. "However, Assumption 2 posits a bounded domain. Does this imply that the gradients and smoothness are also bounded? In reference [7], such boundedness does not appear to be a requirement."
>
> **A1**. Thanks for the question. We make the following explanations.
>
> - **Bounded domain assumption**: The bounded domain assumption does *not* imply the boundedness of gradients and smoothness. For example, considering the online portfolio selection (OPS) problem [1], where the decision $x_t \in \Delta_d$ and the loss function is $f_t(x) = -\ln \langle x , r_t\rangle$ with $r_t \in (0, 1]^d$. In this case, the feasible domain is bounded while the Lipschitzness and smoothness could be arbitrarily large. In fact, in the online learning community, research under the assumption of bounded domain and unbounded Lipschitzness [2], as well as research under the assumption of unbounded domain and bounded Lipschitzness [3], are both studied and usually conducted parallelly.
> - **Requirement on the boundedness**: We clarify that the proposed algorithms in our paper are Lipschitz-adaptive, meaning they do *not* require the upper bounds of Lipschitzness as an algorithmic input, and they can handle unbounded (but finite) Lipschitzness. Regarding reference [7], we believe their method is not Lipschitz-adaptive. In fact, in Section 6 of [7], when introducing the regularizer $\mathcal{R}_t(x)$, an upper bound of Lipschitzness is required to tune the algorithm.
>
> ---
>
> **Q2**. "On page 5, line 203, the paper asserts that the complexity $(O(D\sqrt{V_T} + LD^2))$ aligns with the optimal constant dependency. It would be helpful if the authors could specify where exactly in reference [7] this lower bound is discussed."
>
> **A2**. Sorry for this misleading expression.  We wish to clarify that our result matches the existing upper bound designed for global smoothness [4, 5] of order $O(D\sqrt{V_T} + LD^2)$, in which our result matches the dependence on all terms including $V_T$ and the constants $D$ and $L$. From a lower bound side, it's known that the gradient-variation lower bound for convex functions is $\Omega(\sqrt{V_T})$ [see discussion above Lemma 9 of [7]), but there remains a lack of lower bounds for $D$ and $L$. We will revise the paper to provide more precise expressions. Thank you!
>
> ---
>
> **Q3**. "The clarity and effectiveness of the paper could be improved by condensing the extensive paragraphs that discuss ‘challenges’ ...This would provide readers with a clearer understanding of the paper's contributions to the current body of knowledge."
>
> **A3**. We appreciate your suggestion! In the revised version, we will condense the content and highlight the paper’s contributions. Thank you for highlighting this point for improvement.
>
> ---
>
> **References:**
>
> [1] Universal portfolios, Mathematical Finance 1991.
>
> [2] Lipschitz adaptivity with multiple learning rates in online learning, COLT 2019.
>
> [3] Black-box reductions for parameter-free online learning in Banach spaces, COLT 2018.
>
> [4] Optimistic online mirror descent for bridging stochastic and adversarial online convex optimization, ICML 2023.
>
> [5] Universal online Learning with gradient variations: A multi-layer online ensemble approach, NeurIPS 2023.
>
> [Paper ref 7] (referred to in the paper) Online optimization with gradual variations, COLT 2012.

---

> > ### Comment · Reviewer_ian3 · 2024-08-12
> >
> > Thank you for the authors' reply. I do not have further questions.

---

### Official Review · Reviewer_VaQR · 2024-07-15

**Soundness:** 3
**Presentation:** 3
**Contribution:** 3
**Rating:** 7
**Confidence:** 4

**Summary:**

This paper studies adaptive online learning under the generalized smoothness assumption. The authors proposed optimistic OMD based algorithms that achieves the first gradient variation bounds under this general setting. Under this assumption, they also provide uninveral algorithms which can adapt to different function types, and also algorithms with dynamic regret.

**Strengths:**

1. This paper is the first to study adaptive online learning under generalized smoothness, which is a much milder condition compared to the standard global smoothness that considered by previous work.


2. In section 2, the authors provide optimistic OMD based algorithms with adaptive step size, and prove that it enjoys O(sqrt{V_T}) and O(log V_T/\lambda) regret bounds for convex and strongly convex functions. To my knowledge, these are the first adaptive bounds under the generalized smoothness condition, which is a novel and interesting contribution.

3. The authors also provide a universal algorithm for achieving gradient variation bounds under the generalized smoothness setting, which can adapt to convex and strongly convex functions, without knowing the type of loss functions in advance. Existing universal methods cannot be directly applied due to the unknown Lipschitzness parameters, and the authors address it by using novel techniques such as function-variation-to-gradient-variation conversion.

4. The authors also showed that the proposed methods can be applied to get adaptive bounds in stochastic extended adversarial online learning and the fast-rate games under more general assumptions.

**Weaknesses:**

1. The base learner proposed in Section 2 needs to know the Smooth constant of the previous observed loss functions (at round t-1), which might be difficult to compute if the loss functions are complicated.

2. The proposed universal algorithm requires to query the gradient O(\log T) times per iterations, while most of the existing universal methods only need to query the gradient once.

3. Previous universal algorithms with gradient variation bound can deal with exp-concave functions, while the proposed algorithm can only deal with convex and strongly convex functions.

4. Previous work can also achieve small-loss bound in this setting. It would be great to understand if small-loss bound is achievable under this more general assumption.


Minor comments:

Line 56: sqrt{V}_T: T should be inside the square root;

Line 140: “a finte bound but unknown L” is not assumed in this paper

Line 197: it is more common to use equality for the big-O notation.

**Questions:**

Please see weakness

**Limitations:**

Yes

---

> ### Author Rebuttal · Authors · 2024-08-05
>
> Thank you for your valuable feedback and appreciation of our work! We will respond to your comments below.
>
> ---
>
> **Q1**. "The base learner proposed in Section 2 needs to know the Smooth constant of the previous observed loss functions (at round $t-1$), which might be difficult to compute if the loss functions are complicated." "The proposed universal algorithm requires to query the gradient $O(\log T)$ times per iterations, while most of the existing universal methods only need to query the gradient once."
>
> **A1**. Thanks for the comments. We will respond to these two comments together since the query for gradients and smoothness are coupled.
>
> - **Computation for smoothness**: The computation of smooth constants can be difficult for generalized smoothness. Given that we work on a weaker notion of smoothness in the online setting, it is essential to query the smoothness at each time step to adapt to the optimization trajectory. When specifying $(L_0, L_1)$-smoothness, a representative instance of generalized smoothness often observed in neural network optimization [1, 2], the smoothness constants can be efficiently estimated by the definition $L_t = L_0+L_1*\\| \nabla f_t(x_t)\\|$.
> - **Multiple queries for gradients**: Existing universal methods are developed under global smoothness, where the gradients and smoothness of the combined decision $x_t = \sum_{i=1}^N p_{t,i}x_{t,i}$ can be shared to tune the base-learners. Under the generalized smoothness condition, the smoothness is highly correlated to the optimization trajectory, and its estimation is guaranteed only within a local region. The smoothness at point $x_t$ might be improper to tune the base-learners, which in turn affects the submitted decision $x_{t+1}$, complicating the tuning process. Therefore, at this stage, we have to decouple the analysis from the meta and base levels and query the gradients, which are related to smoothness by function $\ell_t$, multiple times to analyze the base learners separately. Reducing the number of gradient queries is definitely an important future direction. Thanks for pointing this out!
>
> ---
>
> **Q2**. "Previous universal algorithms with gradient variation bound can deal with exp-concave functions, while the proposed algorithm can only deal with convex and strongly convex functions."
>
> **A2**. Thanks for the question. As discussed in Remark 1 and Remark 2 of the paper, dealing with exp-concave functions is particularly challenging for gradient-variation online learning under generalized smoothness. Specifically, at the base level, the challenge arises because the online Newton step (ONS) algorithm seems unable to benefit from arbitrary optimism. This limitation hinders our ability to tune ONS locally by setting the optimism as $M_t = \nabla f_{t-1}(\hat{x}_t)$. Learning with arbitrary optimism for exp-concave functions remains open. At the meta level, the challenge arises when accommodating heterogeneous inputs, which necessitates a specific optimism design in the meta-algorithm. This optimism design cannot leverage the negative terms introduced by the "exp-concave curvature".
>
> ---
>
> **Q3**. "Previous work can also achieve small-loss bound in this setting. It would be great to understand if small-loss bound is achievable under this more general assumption."
>
> **A3**. Thank you for the comment! Yes, the small-loss bound can be achieved in this setting. Specifically, taking convex functions as an example, we can employ SOAL [3] as the base-algorithm and Algorithm 1 in the paper with $m_{t,i} = \langle \nabla f_t(x_t), x_t - x_{t,i}\rangle$  as the meta-algorithm. These combinations can guarantee a regret bound of $O(\sqrt{\sum_{t=1}^T \\|\nabla f_t(x_t)\\|^2})$ for convex functions. The generalized smooth functions also enjoy the self-bounding property that $\\|\nabla f_t(x_t)\\|^2 \lesssim f_t(x_t) - \min_{x} f_t(x)$ [4, Lemma 3.5], which can further convert the obtained regret bound to the small-loss bound with standard arguments. It remains an interesting question whether it is possible to obtain both the small-loss and gradient-variation results simultaneously with one algorithm. In the future, we will investigate this problem.
>
> ---
>
> **Q4**. minor comments regarding typos and formatting issues
>
> **A4**. Thanks for your careful review and helpful suggestions. We will revise and improve our paper presentation accordingly.
>
> ---
>
> **References:**
>
> [1] Why gradient clipping accelerates training: A theoretical justification for adaptivity, ICLR'20.
>
> [2] Robustness to unbounded smoothness of generalized signSGD, NeurIPS'22.
>
> [3] Scale-free online learning, TCS'18.
>
> [4] Convex and non-convex optimization under generalized smoothness, NeurIPS'23.

---

> > ### Comment · Reviewer_VaQR · 2024-08-09
> > **Response**
> >
> > Thank you for the detailed response. I have no further questions at this time.
> >
> > After reading the other reviews, I agree with Reviewer NEvk that Section 4 appears to have been written hastily. It is only 1/3 of a page but contains too much information/results. Additionally, the nonstationary part of the paper is not closely related to the universal results, which are the primary focus of the paper. It would be great if the authors could consider moving this section to the appendix or a journal extension.

---

> > > ### Author Response · Authors · 2024-08-10
> > > **Re: Response**
> > >
> > > Thank you for your comments! We will work on improving the clarity of the paper presentation to ensure that the key ideas and main results are conveyed more clearly.

---

### Decision · Program_Chairs · 2024-09-25

**Decision:**

Accept (poster)

**Comment:**

Typically, regret bounds  in terms of "gradient variation" $\|\nabla f_t(x) -\nabla f_{t-1}(x)\|$  require a smoothness condition. This work  extends this notion to a weaker smoothness condition that has recently been used for analysis of non-convex objectives in deep learning.

Overall the reviewers have a positive opinion  of the  novelty and technical significance of this work, so it is recommended for acceptance.